# Grammar-Based Grounded Lexicon Learning

**Jiayuan Mao**
MIT

**Haoyue Shi**
TTIC

**Jiajun Wu**
Stanford University

**Roger P. Levy**
MIT

**Joshua B. Tenenbaum**
MIT

## Abstract

We present Grammar-Based Grounded Lexicon Learning (G2L2), a lexicalist approach toward learning a compositional and grounded meaning representation of language from grounded data, such as paired images and texts. At the core of G2L2 is a collection of lexicon entries, which map each word to a tuple of a syntactic type and a neuro-symbolic semantic program. For example, the word *shiny* has a syntactic type of *adjective*; its neuro-symbolic semantic program has the *symbolic* form $\lambda x.\mathit{filter}(x, \mathbf{SHINY})$, where the concept $\mathbf{SHINY}$ is associated with a *neural network* embedding, which will be used to classify shiny objects. Given an input sentence, G2L2 first looks up the lexicon entries associated with each token. It then derives the meaning of the sentence as an executable neuro-symbolic program by composing lexical meanings based on syntax. The recovered meaning programs can be executed on grounded inputs. To facilitate learning in an exponentially-growing compositional space, we introduce a joint parsing and expected execution algorithm, which does local marginalization over derivations to reduce the training time. We evaluate G2L2 on two domains: visual reasoning and language-driven navigation. Results show that G2L2 can generalize from small amounts of data to novel compositions of words.

## 1 Introduction

Human language learning suggests several desiderata for machines learning from language. Humans can learn grounded and compositional representations for novel words from few examples. These representations are grounded on contexts, such as visual perception. We also know how these words relate with each other in composing the meaning of a sentence. *Syntax*—the structured, order-sensitive relations among words in a sentence—is crucial in humans' learning and compositional abilities for language. According to *lexicalist* linguistic theories [30, 37, 9], syntactic knowledge involves a small number of highly abstract and potentially universal combinatory rules, together with a large amount of learned information in the lexicon: a rich syntactic type and meaning representation for each word.

Fig. 1 illustrates this idea in a visually grounded language acquisition setup. The language learner looks at a few examples containing the novel word *shiny* (Fig. 1a). They also have a built-in, compact but universal set of combinatory grammar rules (Fig. 1b) that describes how the semantic program of words can be combined based on their syntactic types. The learner can recover the syntactic type of the novel word and its semantic meaining. For example, *shiny* is an adjective and its meaning can be grounded on visually shiny objects in images (Fig. 1c). This representation supports the interpretation of novel sentences in a novel visual context (Fig. 1d).

In this paper, we present Grammar-Based Grounded Lexicon Learning (G2L2), a neuro-symbolic framework for grounded language acquisition. At the core of G2L2 is a collection of grounded lexicon entries. Each lexicon entry maps a word to (i) a syntactic type, and (ii) a neuro-symbolic semantic program. For example, the lexicon entry for the English word *shiny* has a syntactic type of *objset/objset*: it will compose with another constituent of type *objset* on its right, and produces a new

---

Correspondence to Jiayuan Mao: jiayuanm@mit.edu. Project page: http://g2l2.csail.mit.edu.

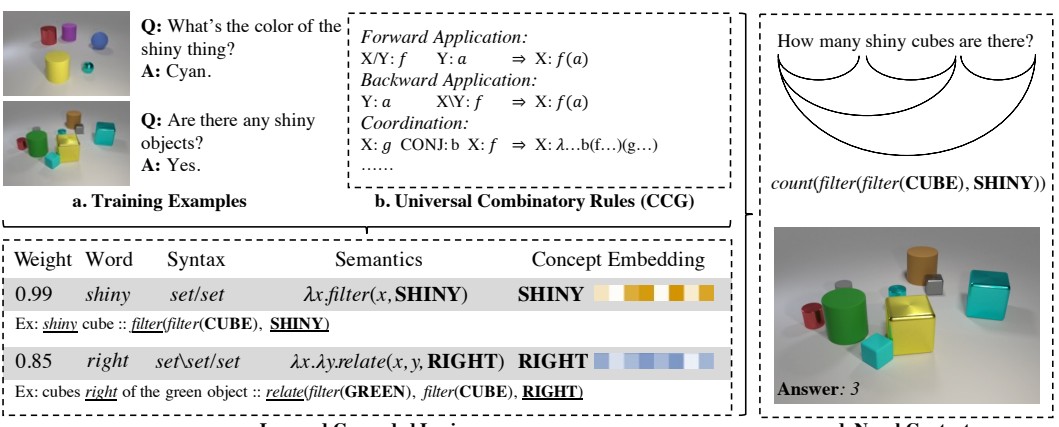

Figure 1: G2L2 learns from grounded language data, for example, by looking at images and reading parallel question–answer pairs. It learns a collection of grounded lexicon entries comprised of weights, syntax types, semantics forms, and optionally, grounded embeddings associated with semantic concepts. These lexicon entries can be used to parse questions into programs.

constituent of syntactic type *objset*. For example, in Fig. 1d, the word *shiny* composes with the word *cube* and yields a new constituent of type *objset*. The neuro-symbolic semantic program for *shiny* has the form $\lambda x.filter(x, \mathbf{SHINY})$, where **SHINY** is a concept *automatically* discovered by G2L2 and associated with a learned vector embedding for classifying shiny objects. G2L2 parses sentences based on these grounded lexicon entries and a small set of combinatory categorial grammar [CCG; 36] rules. Given an input question, G2L2 will lookup the lexicon entries associated with each token, and compose these lexical semantic programs based on their syntactic types.

G2L2 takes a lexicalist approach toward grounded language learning and focuses on data efficiency and compositional generalization to novel contexts. Inspired by lexicalist linguistic theories, but in contrast to neural network-based end-to-end learning, G2L2 uses a compact symbolic grammar to constraint how semantic programs of individual words can be composed, and focuses on learning the lexical representation. This approach brings us strong data efficiency in learning new words, and strong generalization to new word compositions and sentences with more complex structures.

We are interested in jointly learning these neuro-symbolic grounded lexicon entries and the grounding of individual concepts from grounded language data, such as by simultaneously looking at images and reading parallel question–answer pairs. This is particularly challenging because the number of candidate lexicon entry combinations of a sentence grows exponentially with respect to the sentence length. For this reason, previous approaches to lexicon learning have either assumed an expert-annotated set of lexical entries [46] or only attempted to learn at very small scales [15]. We address this combinatory explosion with a novel joint parsing and *expected execution* mechanism, namely CKY-E2, which extends the classic CKY chart parsing algorithm. It performs local marginalization of distributions over sub-programs to make the search process tractable.

In sum, our paper makes three specific contributions. First, we present the neuro-symbolic G2L2 model that learns grounded lexical representations without requiring annotations for the concepts to be learned or partial word meanings; it automatically recovers underlying concepts in the target domain from language and experience with their groundings. Second, we introduce a novel expected execution mechanism for parsing in model training, to facilitate search in the compositional grammar-based space of meanings. Third, through systematic evaluation on two benchmarks, visual reasoning in CLEVR [21] and language-driven navigation in SCAN [25], we show that the lexicalist design of G2L2 enables learning with strong data efficiency and compositional generalization to novel linguistic constructions and deeper linguistic structures.

## 2 Grammar-Based Grounded Lexicon Learning

Our framework, Grammar-Based Grounded Lexicon Learning (G2L2) learns grounded lexicons from cross-modal data, such as paired images and texts. Throughout this section, we will be using the visual reasoning task, specifically visual question answering (VQA) as the example, but the idea itself can be applied to other tasks and domains, such as image captioning and language-driven navigation.

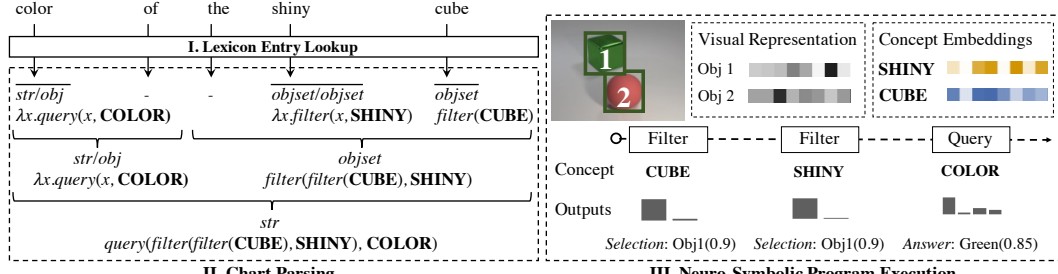

**II. Chart Parsing**     **III. Neuro-Symbolic Program Execution**

Figure 2: G2L2 parses the input sentence into an executable neuro-symbolic program by first (I) lookup the lexicon entry associated with each word, followed by (II) computes the most probable parsing tree and the corresponding tree with a chart parsing algorithm. The derived program can be grounded and executed on an image with a neuro-symbolic reasoning process [28] (III).

G2L2 learns from a collection of VQA data tuples, containing an image, a question, and an answer to the question. In G2L2, each word type $w$ is associated with one or multiple lexical entries, comprised of their syntactic types and semantic programs. Given the input question, G2L2 first looks up the lexicon entries associated with each individual token in the sentence (Fig. 2I). G2L2 then uses a chart parsing algorithm to to derive the programmatic meaning representation of the entire sentence by recursively composing meanings based on syntax (Fig. 2II). To answer the question, we execute the program on the image representation (Fig. 2III). During training, we compare the answer derived from the model with the groundtruth answer to form the supervision for the entire system. No additional supervision, such as lexicon entries for certain words or concept labels, is needed.

## 2.1 Grounded Lexicon

At a high-level, G2L2 follows the combinatory categorical grammar [CCG; 36] formalism to maintain lexicon entries and parse sentences. Illustrated in Fig. 3, Each word $w$ (e.g., *shiny*) is associated with one or multiple entries. Each entry $e_w^{(i)}$ is a tuple comprised of a syntax type $syn_w^{(i)}$ (e.g., *objset/objset*), and a semantic meaning form $sem_w^{(i)}$ (e.g., $\lambda x.filter(x, \textbf{SHINY})$). $sem_w^{(i)}$ is a symbolic program represented in a typed domain-specific language (DSL) and can be executed on the input image. Some programs contain concepts (in this case, **SHINY**) that can be visually grounded.

**Typed domain specific language.** G2L2 uses a DSL to represent word meanings. For the visual reasoning domain, we use the CLEVR DSL [21]. It contains object-level operations such as selecting all objects having a particular attribute (e.g., the shiny objects) or select all objects having a specific relationship with a certain object (e.g., the objects left of the cube). It also supports functions that respond to user queries,

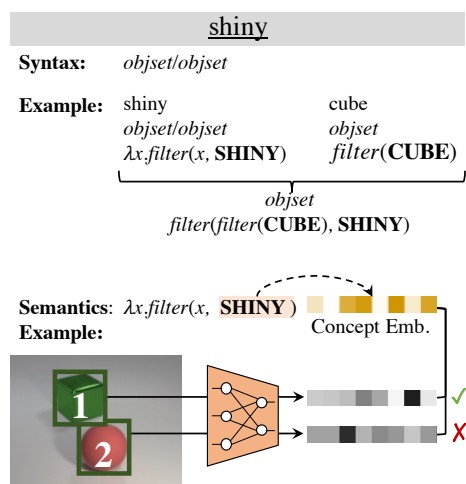

Figure 3: Each word is associated with a grounded lexicon, comprised of its syntactic type and a neuro-symbolic semantic program.

such as counting the number of objects or query a specific attribute (e.g., shape) of an object. The language is typed: most functions takes a set of objects or a single object as their inputs, and produce another set of objects. For example, the operation *filter* has the signature $filter(objset, concept) \rightarrow objset$ and returns all objects that have *concept* (e.g., all *shiny* objects) in the input set.

**Syntactic types.** There are two types of syntactic types in G2L2: primitive and complex.[*] The primitive types are defined in the typed domain specific language (e.g., *objset*, *int*). A complex type, denoted as X/Y or X\Y, is a functor type that takes an argument of type Y and returns an object of type X. The direction of the slash indicates word order: for X/Y, the argument Y must appear on the right, whereas in X\Y, it must appear on the left. Note that X and Y can themselves be complex types,

---

[*]In some domains we also use conjunctions (*CONJ*) in the coordination rule.

which allows us to define functor types with multiple arguments, such as (X\Y)/Z, or even functors with functor arguments (e.g., (X\Y)/(Z/Z)).

In G2L2, the semantic type of a word meaning (in the DSL) together with a set of directional and ordering settings for its arguments (reflecting how the word and its arguments should be linearized in text) uniquely determines the word's syntactic type. For example, the syntactic type for word *shiny* is *objset/objset*. It first states that *shiny* acts as a function in meaning composition, which takes a subprogram that outputs a set of objects (e.g., *filter*(**CUBE**)) as its argument, and produces anew program whose output is also a set of objects, in this case, *filter*(*filter*(**CUBE**), **SHINY**) Second, it states the direction of the argument, which should come from its right.

**Neuro-symbolic programs.** Some functions in the DSL involves concepts that will be grounded in other modalities, such as the visual appearance of an object and their spatial relationships. Taking the function *filter* as an example, its secondary argument *concept* should be associated with the visual representation of objects. In G2L2, the meaning of each lexicon entry may involve one more constants (called "concepts") that are grounded on other modalities, possibly via deep neural embeddings. In the case of *shiny*: $\lambda x.filter(x, \textbf{SHINY})$. The concept **SHINY** is associated with a vector embedding in a joint visual-semantic embedding space, following Kiros et al. [24]. During program execution, we will be comparing the embedding of concept **SHINY** with object embeddings extracted from the input image, to filter out all *shiny* objects.

**Lexicon learning.** G2L2 learns lexicon entries in the following three steps. (i) First, we enumerate all possible semantic meaning programs derived from the DSL. For example, in the visual reasoning domain, a candidate program is $\lambda x.filter(x, \textbf{?})$, where **?** denotes a concept argument. When we try to associate this lexicon entry to the word *shiny*, the program is instantiated as $\lambda x.filter(x, \textbf{SHINY})$, where **SHINY** is a new concept associated with a vector embedding. Typically, we set a maximum number of arguments for each program and constrain its depth. We explain how we set these hyperparameters for different domains in the supplementary material. (ii) Next, for programs that have a primitive type, we use its semantic type as the syntactic type (e.g., *objset*). For programs that are functions with arguments, we enumerate possible argument ordering of the arguments. For example, the program $\lambda x.filter(x, \textbf{SHINY})$ has two candidate syntactic types: *objset/objset* (the argument is on its right in language) and *objset\objset* (the argument is on its left). (iii) Finally, we associate each candidate lexicon entry with a learnable scalar weight $\tau(\cdot)$. It is typical for a single word having tens or hundreds of candidate entries, and we optimize these lexicon entry weights in the training process. In practice, we assume no lexical ambiguity, i.e., *each word type has only one lexical entry*. Thus, the ambiguity of parsing only comes from different syntactic derivation orders for the same lexical entries. This also allows us to prune lexicon entries that do not lead to successful derivations during training.

## 2.2 Program Execution

Any fully grounded programs (i.e., programs without unbound arguments) can be executed based on the image representation. We implement the Neuro-Symbolic Concept Learner [NS-CL; 28] as our differentiable program executor, which consists of a collection of deterministic functional modules to realize the operations in the DSL. NS-CL represents execution results in a "soft" manner: in the visual reasoning domain, a set of objects is represented as a vector mask $m$ of length $N$, where $N$ is the number of objects in the scene. Each element, $m_i$ can be interpreted as the probability that object $i$ is in the set. For example, the operation $\lambda x.filter(x, \textbf{SHINY})$ receives an input mask $m$ and produces a mask $m'$ that selects all shiny objects in the input set. The computation has two steps: (i) compare the vector embedding of concept **SHINY** with all objects in the scene to obtain a mask $m^{(\textbf{SHINY})}$, denoting the probability of each object being *shiny*; (ii) compute the element-wise multiplication $m' = m \odot m^{\textbf{SHINY}}$, which can be further used as the input to other functions. In NS-CL, the execution result of any program is fully differentiable w.r.t. the input image representation and concept embeddings (e.g., **SHINY**).

## 2.3 Joint Chart Parsing and Expected Execution (CKY-E2)

G2L2 extends a standard dynamic programming algorithm for chart parsing (i.e., the CKY algorithm [22, 45, 10]) to compose sentence meaning from lexical meaning forms, based on syntax. Denote $w_i$ as the input word sequence. $e_i^j$ the $j$-th lexicon entry associated with word $w_i$, and $\tau(e_i^j)$ the corresponding weight. Consider all possible derivation of the question $\{derivation_k\}$, $k = 1, 2, \ldots$. We define the following context-free probability distribution of derivations as:

**Algorithm 1** The CKY-E$^2$ algorithm.

**Input:** $w_i$: the input sentence; $L$: sentence length; $e_i^j$: the $j$-th lexicon entry associated with word $w_i$; $\tau(e_i^j)$: lexicon weights.
**Output:** $exe_k$ the execution result of the all possible derivations and their weights $\tau(exe_k)$.

1: **for** $i \leftarrow 0$ to $L-1$ **do**
2:     Initialize $dp[i, i+1]$ with lexicon entries $e_i^*$ and weights $\tau(e_i^*)$
3: **end for**
4: **for** $length \leftarrow 1$ to $L$ **do**
5:     **for** $left \leftarrow 0$ to $L - length$ **do**
6:         $right \leftarrow left + length$
7:         $dp[left, right] \leftarrow$ empty list
8:         **for** $k \leftarrow left + 1$ to $right - 1$ **do**
9:             Try to combine nodes in $dp[left, k]$ and $dp[k, right]$
10:            Append successful combination to $dp[left, right]$
11:         **end for**
12:         EXPECTEDEXECUTION($dp[left, right]$)
13:     **end for**
14: **end for**
15: **procedure** EXPECTEDEXECUTION($a$: a list of derivations)
16:     **while** $\exists x, y \in a$ are identical except for subtrees of the same type **do**
17:         Create $z$ from $x$ and $y$ by computing the expected execution results for non-identical subtrees
18:         $\tau(z) \leftarrow \tau(x) + \tau(y)$
19:         Replace $x$ and $y$ in $a$ with $z$
20:     **end while**
21: **end procedure**

Span: right of the green object
Candidate 1:
$\lambda x.relate(x, filter(\textbf{GREEN}), \textbf{RIGHT})$

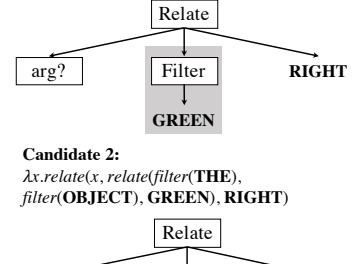

Candidate 2:
$\lambda x.relate(x, relate(filter(\textbf{THE}), filter(\textbf{OBJECT}), \textbf{GREEN}), \textbf{RIGHT})$

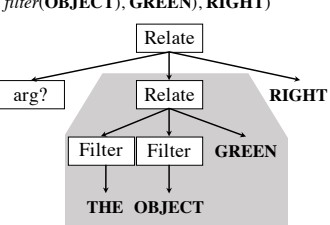

Figure 4: An illustrative example of two semantic programs that can be merged by computing the expected execution results of two subtrees (highlighted in gray). Both subtrees outputs a vector of scores indicating the objects being selected.

$$p(derivation_k) \propto \exp\left(\sum_{e \in derivation_k} \tau(e)\right).$$

That is, the probability is exponentially proportional to the total weights $\tau(e)$ of all lexicon entries $e \in derivation_k$ used by the specific derivation.

A straightforward implementation to support joint learning of lexicon weights $\tau$ and neural modules (e.g., $filter(x, \textbf{SHINY})$), is to simply execute all possible derivations on the input image, and compare the answer with the groundtruth. However, the number of possible derivations grows exponentially as the question length, making such computation intractable. For example, in SCAN [25], each word has 178 candidate lexicons, and the number of lexicon combination of a sentence with 5 words will be $178^5 \approx 10^{11}$. To address this issue, we introduce the idea of expected execution, which essentially computes the "expected" execution result of all possible derivations. We further accelerate this process by taking local marginalization.

Our CKY-E2 algorithm is illustrated in Algorithm. 1. It processes all spans $[left, right)$ sequentially ordered by their length. The composition for derivations of $[left, right)$ has two stages. First, it enumerates possible split point $k$ and tries to combine the derivation of $[left, k)$ and $[k, right)$. This step is identical to the standard CKY parsing algorithm. Next, if there are two derivations $x$ and $y$ of span $[i, j)$, whose program structures are identical except for subtrees that can be partially evaluated (i.e., does not contain any unbounded arguments), we will compress these two derivations into one, by marginalizing the execution result for that subtree.

See the example from Fig. 4. Two programs have the identical structure, except for the second argument to the outer-most *relate* operation. However, these sub-trees, highlighted in gray, can be partially evaluated on the input image, and both of them output a vector of scores indicating the objects being selected. Denote $\tau_1$ and $\tau_2$ as the weight associated with two derivations, and $v_1$ and $v_2$ the partial evaluation results (vectors) for two subtrees. We will replace these two candidate meaning form with $z$:

$$z := \lambda x.relate(x, v', \textbf{RIGHT}), \quad \text{where } v' := \frac{\tau_1 v_1 + \tau_2 v_2}{\tau_1 + \tau_2}, \tau(z) := \tau_1 + \tau_2.$$

We provide additional running examples of the algorithm in the supplementary material.

**Complexity.** Intuitively, once we have determined the semantics of a constituents in the question, the actual concrete meaning form of the derivation does not matter for future program execution, if the meaning form can already be partially evaluated on the input image. This joint parsing and expected execution procedure significantly reduces the exponential space of possible parsing to a polynomial space w.r.t. the number of possible program layouts that can not be partially evaluated, which, in practice, is small. The complexity of CKY-E2 is polynomial with respect to the length $L$ of the sentence, and $M$ the number of candidate lexicon entries. More specifically, $O(L^3 M)$, where $O(L^3)$ comes from the chart parsing algorithm, and the number of derivations after the expected execution procedure is $O(M)$. This result is obtained by viewing the maximum arity for functor types being a constant (e.g., 2). Intuitively, for each span, all possible derivations associated with this span can be grouped into 4 categories: derivations of a primitive type, derivations of a 1-ary functor type, derivations of a 2-ary functor type, and derivations of a 2-ary functor type, with one argument binded. All these numbers grow linearly w.r.t. $M$. For detailed analysis please refer to our supplementary material.

**Correctness.** One can theoretically prove that, if all operations in the program layout are commutative with the expectation operator, i.e., if $\mathbb{E}\left(\left[f\left(x\right)\right] = f\left(\mathbb{E}\left[x\right]\right)\right.$, our CKY-E2 produces exact computation of the expected execution result. These operations include, tensor addition, multiplication (if tensors are independent), and concatenation, which cover most of the computation we will do in neuro-symbolic program execution. For example, for *filter*, taking the expectation over different inputs before doing the filtering is the same as taking the expectation over the filter results of different inputs. However, there are operations such as quantifiers whose semantics are not commutative with the expectation operator. In practice, it is possible to still use the expected expectation framework to approximate. We leave the application of other approximated inference techniques as future work. We provide proofs and its connections with other formalisms in the supplementary material.

## 2.4 Learning

Our model, G2L2, can be trained in an end-to-end manner, by looking at images and reading paired questions and answers. We denote $\ell$ as a loss function that compares the output of a program execution (e.g., a probability distribution over possible answers) and the groundtruth. More precisely, given all possible derivations $derivation_k$, the image representation $I$, the answer $A$, and the executor $\mathcal{E}(\cdot, I)$, we optimize all parameters by minimizing the loss $\mathcal{L}$:

$$\mathcal{L} = \sum_k \left(p(derivation_k) \cdot \ell\left(\mathcal{E}(derivation_k, I), A\right)\right).$$

In practice, we use gradient-based optimization for both the neural network weights in concept grounding modules and the lexicon weights $\tau$.

## 3 Experiment

We evaluate G2L2 on two domains: visual reasoning in CLEVR [21] and language-driven navigation in SCAN [25]. Beyond the grounding accuracy, we also evaluate the compositional generalizability and data efficiency, comparing G2L2 with end-to-end neural models and modular neural networks.

### 3.1 Visual Reasoning

We first evaluate G2L2 on the visual reasoning tasks in the CLEVR domain [21], where the task is to reason and answer questions about images. In our study, we use a subset of CLEVR dataset, which does not include sentences that involve coreference resolution, and words with multiple meanings in different contexts. We add additional information on how we filter the dataset in the supplementary.

**Setup.** Instead of using manually defined heuristics for curriculum learning or self-paced learning as in previous works [28, 26], we employ a curriculum learning setup that is simply based on sentence length: we gradually add longer sentences into the training set. This helps the model to learn basic words from very short sentences (6 words), and use the acquired lexicon to facilitate learning longer sentences (20 words). Since CLEVR does not provide test set annotations, for all models, we held out 10% of the training data for model development and test them on the CLEVR validation split.

**Baselines.** We compare G2L2 with 4 baselines. (1) MAC [19] is an end-to-end approach based on attention. (2) TbD-Net [29] uses a pre-trained semantic parser to parse the question into a symbolic

| Model | Prog? | Concept? | Standard | | Compositional Generalization | | | Depth |
|---|---|---|---|---|---|---|---|---|
| | | | 10% | 100% | *purple* | *right of* | *count* | |
| MAC [19] | N/N | N/N | 85.39 | 98.61 | 97.14 | 90.85 | 54.87 | 77.40 |
| TbD-Net [29] | Y/Y | N/N | 44.52 | 98.04 | 89.57 | 49.92 | 63.37 | 53.13 |
| NS-VQA [44] | Y/Y | Y/Y | **98.57** | 98.57 | 95.52 | **99.80** | 81.81 | 50.45 |
| NS-CL [28] | Y/N | Y/N | 98.51 | **98.91** | **98.02** | 99.01 | 18.88 | 81.60 |
| G2L2 (ours) | Y/N | Y/N | 98.11 | 98.25 | 97.82 | 98.59 | **96.76** | **98.49** |

Table 1: Accuracy on the CLEVR dataset. Our model achieves a comparable results with state-of-the-art approaches on the standard training-testing split. It significantly outperforms all baselines on generalization to novel word compositions and to sentences with deeper structures. The best number in each column is bolded. The second column indicates whether the model uses program-based representation of question meaning and whether it needs program annotation for training questions. The third column indicates whether the model explicitly models individual concepts and whether it needs concept annotation for objects during training.

program, and executes the program with a neural module network [5]. (3) Similarly, NS-VQA [44] also parses the question into a symbolic program. It also extracts an abstract scene representation with pre-trained neural recognition models [17]. It executes the program based on the abstract scene representation. Both of the approaches require additional supervision for training the semantic parser, and NS-VQA requires additional annotation for training the visual recognition model. (4) NS-CL [28] jointly learns a neural semantic parser and concept embeddings by looking at images and reading paired questions and answers. It requires the annotation for all concepts in the domain (e.g., colors and shapes). In contrast, G2L2 can *automatically* discover visual concepts from texts.

**Results.** Table 1 summarizes the results. We consider any model that performs in the 95–100 range to have more or less solved the task. Small differences in numeric scores in this range, such as the fact that NS-CL outperforms our model on the "purple" generalization task by 0.2%, are less important than the fact that our model far outperforms all competitors on "count" compositional generalization and the "depth" generalization task, both of which all competitor models are far from solving.

We first compare different models on the **standard** training-testing split. We train different models with either 10% or 100% of the training data and evaluate them on the validation set. Our model achieves a comparable performance in terms of its accuracy and data efficiency.

Next, we systematically build three **compositional generalization** test splits: *purple*, *right of*, and *count*. The detailed setups and examples for these splits are provided in the supplementary. Essentially, we remove 90% of the sentences containing the word *purple*, the phrase *right*, and *counting operations*, such as *how many ...?* and *what number of ...?*. We only keep sentences up to a certain length (6 for purple, 11 for right, and 8 for count). We make sure that each use case of these words appear in training questions. After training, we test these models on the validation set with questions containing these words. Overall, our model G2L2 outperforms all baselines on all three generalization splits. In particular, it significantly outperforms other methods on the *count* split. The *count* split is hard for other method because it requires model to generalize to sentences with deeper structures, for example, from "*how many red objects are there?*" to "*how many red objects are right of the cube?*" Note that, during training, all models have seen example use of similar structures such as "*what's the shape of the red object*" and "*what's the shape of the red object right of the cube?*"

Finally, we test generalization to sentences with deeper structures (**depth**). Specifically, we define the "hop number" of a question as the number of intermediate objects being referred to in order to locate the target object. For example, the "hop number" of the question "*how many red objects are right of the cube?*" is 1. We train different models on 0-hop and 1-hop questions and test them on 2-hop questions. Our model strongly outperforms all baselines.

The results on the **compositional generalization** and **depth** splits yield two conclusions. First, disentangling grounded concept learning (associating words onto visual appearances) and reasoning (e.g., filtering or counting subsets of objects in a given scene) improves data efficiency and generalization. On CLEVR, neuro-symbolic approaches that separately identify concepts and perform explicit reasoning (NS-VQA, NS-CL and G2L2) consistently generalize better than approaches that do not (MAC, TbD). The comparison between TbD and NS-VQA is informative: TbD fails on the "right of" task even in the case where the semantic parser is providing correct programs, while NS-VQA,

| Model | Simple | | Compositional Generalization | | Length |
|---|---|---|---|---|---|
| | 10% | 100% | *jump* | *around right* | |
| seq2seq [38] | 0.93±0.05 | 0.99±0.01 | 0.00±0.00[†] | 0.00±0.00[†] | 0.15±0.02 |
| Transformer [42] | 0.71±0.24 | 0.78±0.11 | 0.00±0.00 | 0.10±0.08 | 0.02±0.01 |
| GECA [3] | 0.99±0.00 | 0.98±0.01 | 0.87±0.05[†] | 0.82±0.11[†] | 0.15±0.02 |
| WordDrop [16]* | 0.56±0.02 | 0.62±0.02 | 0.52±0.02 | 0.70±0.06 | 0.18±0.01 |
| SwitchOut [43]* | 0.99±0.01 | 0.99±0.01 | 0.98±0.02 | 0.97±0.02 | 0.17±0.02 |
| SeqMix [16]* | – | – | 0.98[‡] | 0.89[‡] | – |
| recomb-2 [2] | – | – | 0.88±0.07[†] | 0.82±0.08[†] | – |
| G2L2 (ours) | **1.00**±0.00 | **1.00**±0.00 | **1.00**±0.00 | **1.00**±0.00 | **1.00**±0.00 |

Table 2: Accuracy on the SCAN dataset, averaged across 10 valid runs when applicable, ± denotes standard deviation. The best number in each column is bolded. †: results taken from [2]; ‡: results taken from [16]. Both paper have only presented results on the compositional generalization split. ∗: applied after GECA. The results for GECA are based on the released implementation by the authors. All the models are selected with respect to the accuracy on the training set.

which uses the same parser but explicitly represents compositional symbolic concepts for reasoning, succeeds in this task. Crucially, of the three neuro-symbolic methods, G2L2 achieves strong performance with less domain-specific knowledge than other methods: NS-VQA needs groundtruth programs; NS-CL needs the concept vocabulary; G2L2 requires neither. Second, our model is the only one to perform well on the hardest "out-of-sample" generalization tests: holding out "count" and generalizing to deeper embeddings. The other, easier generalization tests all have close neighbors in the training set, differing by just one word. In contrast, the length, depth and "count" tests require generalizing to sentences that differ in multiple words from any training example. They appear to require – or at least benefit especially well from – G2L2 's lexical-grammatical approach to capturing meaning of complex utterances, with explicit constituent-level (as opposed to simply word-level) composition. We also provide in-depth analysis for the behavior of different semantic parsing models in the supplementary material.

### 3.2 Language-driven Navigation

The second domain we consider is language-driven navigation. We evaluate models on the SCAN dataset [25]: a collection of sentence and navigational action sequence pairs. There are 6 primitive actions: *jump*, *look*, *walk*, *run*, *lturn*, and *rturn*, where an instruction *turn left twice and run* will be translated to *lturn lturn run*. All instructions are generated from a finite context-free grammar, so that we can systematically construct train-test splits for different types of compositional generalizations.

**Setup.** We use a string-editing domain-specific language (DSL) for modeling the meaning of words in the SCAN dataset, of which the details can be found in the supplementary material. At a high level, the model supports three primitive operations: constructing a new constant string (consisting of primitive operations), concatenating two strings, and repeating the input string for a number of times.

For G2L2, we generate candidate lexicons by enumerating functions in the string-editing DSL with up to 2 arguments and the function body has a maximum depth of 3. We also allow at most one of the argument being functor-typed, for example, $V\backslash V/(V\backslash V)$. To handle parsing ambiguities, we use two primitive syntax types $S$ and $V$, while both of them are associated with the semantic type of *string*. In total, we have 178 candidate lexicon entries for each word.

**Baselines.** We compare G2L2 to seven baselines. (1) Seq2seq [38] trains an LSTM-based encoder-decoder model. We follow the hyperparameter setups of [25]. (2) Transformer [42] is a 4-head Transformer-based autoregressive seq2seq model. We tuned the hidden size (i.e., the dimension of intermediate token representations) within {100, 200, 400}, as well as the number of layers (for both the encoder and the decoder) from {2, 4, 8}. Other methods are based on different data augmentation schemes for training a LSTM seq2seq model. Specifically, (3) GECA augments the original training splits using heuristic span recombination rules; (4) WordDrop [16] performs random dropout for input sequence (while keeping the same label); (5) similarly, SwitchOut [43] randomly replaces an input token with a random token from the vocabulary; (6) SeqMix [16] uses soft augmentation techniques following [47], which composes an "weighted average" of different input sequences; (7) recomb-2 [2] learns recombination and resampling rules for augmentation.

**Results.** We compare different models on three train-test splits. In **Simple**, the training and test instructions are drawn from the same distribution. We compare the data efficiency of various models

by using either 10% or 100% of the training data, and test them on the same test split. While all models can achieve a nearly-perfect accuracy with 100% training data, our model G2L2 shows advantage with only a small amount of data. Next, in **Compositional**, we have held out the sentences containing certain phrases, such as *jump* and *around right*. For these held-out phrases, only valid non-contextual examples containing them (i.e., *jump* in isolation and no example for *around right*) are available during training. During test, algorithms need to make systematical generalization of these phrases in novel contexts. Finally, in **Length**, all training examples have the action length less than or equal to 22, while that of a test example is up to 48. Our model consistently reach perfect performance in all considered settings, even on the cross-length generalization task where GECA does not help improve performance. These results are consistent with the conclusions we derived on the CLEVR dataset. Specifically, data-augmentation techniques for SCAN can solve simple generalization tests (e.g., *jump*, where tests all have close neighbors in the training set, differing by just one word) but not the hard ones (e.g., *length*, where test sentences can different in multiple words from any training examples).

**Cases study.** G2L2 is expressive enough to achieve perfect accuracy on the SCAN dataset: there exists a set of lexicon entries which matches the groundtruth in SCAN. However, our learning algorithm does not always converge on the correct lexicon, but when it fails, the failure can be identified based on training-set accuracy. So, we perform model selection based on the training accuracy for G2L2: after a sufficient number of epochs, if the model hasn't reached perfect accuracy (100%), we re-initialize the weights and train the model again. Our results show that, among 100 times of training, the model reaches 100% accuracy 74% of the time. For runs that don't have 100% accuracy, the average performance is 0.94.

Since G2L2 directly learns human-interpretable lexicon entries associated with each individual words, we can further inspect the failure cases made by it when the training accuracy does not converge to 0. We find that the most significant failure mode is the word *and* (e.g., *jump and run*) and *after* (e.g., *jump after run*). Both of them are treated as connectives in SCAN. Sometimes G2L2 fails to pick the syntax type *S\V/V* over the type *V\V/V*. The entry *V\V/V* will succeed in parsing most cases (e.g., *jump and run*), except that it will introduce ambiguous parsing for sentences such as "*jump and run twice*": *jump and run twice vs. jump and run twice*. Based on the definition of the SCAN, only the first derivation is valid. In contrast, using *S\V/V* resolves this ambiguity. Depending on the weight initialization and the example presentation order, G2L2 sometimes get stuck at the local optima of *V\V/V*. However, we can easily identify this by the training accuracies—G2L2 is able to reach perfect performance on all considered splits by simply retraining with another random seed, therefore, we only select those with 100% training accuracy as valid models.

## 4    Related Work

**Lexicalist theories.** The lexicalist theories of syntax [30, 37, 9] propose that 1) the key syntactic principles by which words and phrases combine are extremely simple and general, and 2) nearly all of the complexity in syntax can be attributed to rich and detailed lexical entries for the words in the language. For example, whereas the relationship between the active and passive voice, e.g., "Kim saw a balloon" versus "A balloon was seen by Kim", was treated in pre-lexicalist theories as a special syntactic rule converting between the sentences, in lexicalist theories this relationship derives simply from the knowledge that the passive participle for the verb "see" is "seen," which interacts with knowledge of other words to make both the active and passive forms of the sentence possible. In lexicalist theories, the problem for the language learner thus becomes a problem of learning the words in the language, not a problem of learning numerous abstract rule schemas. The combinatory categorial grammar [CCG; 36] framework we use is a well-established example of a lexicalist theory: there is a universal inventory of just three combinatory rules (Fig. 1a), but those rules can only be applied once richly specified lexical entries are learned for the words in a sentence. We believe that this lexicalist-theory approach is a particularly good fit to the problem of grounded language learning: the visual context provides clues to the word's meaning, and the word's grammatical behavior is tied closely to this meaning, making learning efficient.

**Compositional generalization in NLP.** Improving the compositional generalization of natrual language processing (NLP) systems have drawn great attention in recent years [8]. Most of the recent approaches towards this goal are mostly built on deep learning-based models. There are two representative approaches: building structured neural networks with explicit phrase-based structures or segments [35, 49, 39, 32]; and using data augmentation techniques [3, 16, 2]. However, these

approaches either rely on additional annotation or pretrained models for phrase structure inference or require domain-specific heuristics in data augmentation. In contrast to both approaches, we propose to use combinatory grammar rules to constrain the learning of word meanings and how they compose.

**Neural latent trees.** CKY-E2 is in spirit related to recent work using CKY-style modules for inducing latent trees. However, our model is fundamentally different from works on unsupervised constituency parsing [23, 34] which use the CKY algorithm for inference over scalar span scores and those compute span representation vectors with CKY-style algorithms [27, 11, *inter alia*]. Our key contribution is to introduce the expected execution mechanism, where each span is associated with weighted, compressed programs. Beyond enumerating all possible parsing trees as in [27], G2L2 considers all possible programs associated with each span. Our expected execution procedure works for different types (object set, integer, etc.) and even functor types. This makes our approximation exact for linear cases and has polynomial complexity.

**Grammar-based grounded language learning.** There have also been approaches for learning grammatical structures from grounded texts [33, 48, 20, 7, 31, 41]. However, these approaches either rely on pre-defined lexicon entries [7], or only focus on inducing syntactic structures such as phrase-structure grammar [33]. Different from them, G2L2 jointly learns the syntactic types, semantic programs, and concept grounding, only based on a small set of combinatory grammar rules.

Grammar-based and grounded language learning have also been studied in linguistics, with related work to ours studying on how humans use grammar as constraints in learning meaning [37] and how learning syntactic rules and semantic meanings in language bootstrap each other[1, 40]. However, most previous computational models have focused only on explaining small-scale lab experiments and do not address grounding in visual perception [13, 15]. In contrast, G2L2 is a neuro-symbolic model that integrates the combinatory categorial grammar formalism [36] with joint perceptual learning and concept learning, to directly learn meanings from images and texts.

**Neuro-symbolic models for language grounding.** Integrating symbolic structures such as programs and neural networks has shown success in modeling compositional queries in various domains, including image and video reasoning [18, 29], knowledge base query [4], and robotic planning [6]. In this paper, we use symbolic domain-specific languages with neural network embeddings for visual reasoning in images and navigation sequence generation, following NS-CL [28]. However, in contrasts to using neural network-based semantic parser as in the aforementioned papers, our model G2L2 focuses on learning grammar-based lexicon for compositional generalization in linguistic structures, such as novel word composition.

## 5 Conclusion and Discussion

In this paper, we have presented G2L2, a lexicalist approach towards learning compositional and grounded meaning of words. G2L2 builts in a compact but potentially universal set of combinatory grammar rules and learns grounded lexicon entries from a collection of sentences and their grounded meaning, without any human annotated lexicon entries. The lexicon entries represent the semantic type of the word, the ordering settings for its arguments, as well as the grounding of concepts in its semantic program. To facilitate lexicon entry induction in an exponentially-growing space, we introduced CKY-E2 for joint chart parsing and *expected execution*.

Through systematical evaluation on both visual reasoning and language-driven navigation domains, we demonstrate the data efficiency and compositional generalization capability G2L2, and its general applicability in different domains. The design of G2L2 suggests several research directions. First, in G2L2 we have made strong assumptions on the context-independence of the lexicon entry as well as the application of grammar rules, the handling of linguistic ambiguities and pragmatics needs further exploration [14]. Second, meta-learning models that can leverage learned words to bootstrap the learning of novel words, such as syntactic bootstrapping [15], is a meaningful direction. Finally, future work may consider integrating G2L2 with program-synthesis algorithms [12] for learning of more generic and complex semantic programs.

**Broader impact.** The ideas and techniques in this paper can be potentially used for building machine systems that can better understand the queries and instructions made by humans. We hope researchers and developers can build systems for social goods based on our paper. Meanwhile, we are aware of the ethical issues and concerns that may arise in the actual deployment of such systems, particularly biases in language and their grounding. The strong interpretability of the syntactic types and semantic programs learned by our model can be used in efforts to reduce such biases.

**Acknowledgements.** This work is in part supported by ONR MURI N00014-16-1-2007, the Center for Brain, Minds, and Machines (CBMM, funded by NSF STC award CCF-1231216), the MIT Quest for Intelligence, Stanford Institute for Human-Centered AI (HAI), Google, MIT–IBM AI Lab, Samsung GRO, and ADI. Any opinions, findings, and conclusions or recommendations expressed in this material are those of the authors and do not necessarily reflect the views of our sponsors.

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
