# Supplementary Material for
# Grammar-Based Grounded Lexicon Learning

**Jiayuan Mao**
MIT

**Haoyue Shi**
TTIC

**Jiajun Wu**
Stanford University

**Roger P. Levy**
MIT

**Joshua B. Tenenbaum**
MIT

In the supplementary material, we describe the domain specific languages used in our experiments (Section 1), demonstrate how the proposed CKY-E2 method works by a concrete example (Section 2.1), show formal properties of CKY-E2 (Section 2.2), present dataset setups and analyze model behaviors (Section 3), and list environmental details for experiments (Section **??**).

## 1   Domain Specific Languages and Neuro-Symbolic Reasoning

In this section, we will present and discuss the domain-specific languages (DSLs) we use for two domains: visual reasoning and language-guided navigation. We will further introduce the neuro-symbolic module we have designed for executing programs in these two domains. Overall, each DSL contains a set of types and a set of deterministic modules that have been manually designed for realizing necessary operations in these domains. However, in contrast to realizing them as we do in standard programming languages (with for-loops and if-conditions), we will be using tensor operations (e.g., tensor additions and multiplications) to realize them so that the output of each program is differentiable with respect to all of its inputs.

### 1.1   Visual Reasoning DSL

Our domain-specific language (DSL) for the visual reasoning domain is based on the CLEVR DSL introduced in [6], and the neuro-symbolic realization of each functional module is slightly modified from the Neuro-Symbolic Concept Learner [NS-CL; 8]. We refer readers to the original papers for a detailed introduction to the DSL and neuro-symbolic program execution. Here we only highlight the key aspects of our language and its neuro-symbolic realization, and discuss the difference between our implementation and the ones in original papers.

Our visual reasoning DSL is a subset of CLEVR, containing 6 types and 8 primitive operations. Table 1 illustrates all 6 types and how they are internally represented in neuro-symbolic execution.

Table 2 further shows all operations in the DSL. There are two main differences between the DSL used by G2L2 and the original CLEVR DSL. First, we have removed the *unique* operation, whose semantic meaning was to return the single object in a set of objects. For example, it can be used to represent the meaning of word "*the*" in "*the red object*", in which the semantic program of "*red object*" yields a set of red objects and the semantic program of "*the*" selects the unique object in that set. However, the meaning of "*the*" may have slightly different semantic type in different contexts, for example, "*what is the color of ...*". Since this has violated our assumption about each word having only one lexicon entry, we choose to remove this operation to simplify the learning problem. Meanwhile, to handle the "uniqueness" of the object being referred to, in our realization of related operations, such as *relate* and *query*, we will implicitly choose the unique object being referred to, which we will detail in the following paragraphs.

**Object-centric scene representation.**   In our visual reasoning domain, we have assumed access to a pre-trained object-detector that generates a list of bounding boxes of objects in the scene. In our implementation, following Mao et al. [8], we use a pre-trained Mask R-CNN [4] to generate bounding boxes for each object proposal. These bounding boxes, paired with the original image, are

35th Conference on Neural Information Processing Systems (NeurIPS 2021).

| Type | Note | Representation |
|---|---|---|
| ObjConcept | Object-level concepts. | An embedding vector. |
| Attribute | Object-level attributes. | A vector of length $K_{obj}$, where $K_{obj}$ is the number of |
| RelConcept | Relational concepts. | An embedding vector. |
| ObjectSet | A set of objects in the scene. | A vector $\mathbf{m}$ of length $N$, where $N$ is the number of objects in the scene. Each entry $\mathbf{m}_i$ is a real value in $[0, 1]$, which can be interpreted as the probability that object $i$ is in this set. |
| Integer | An integer. | A single non-negative real value, which can be interpreted as the "expected" value of this integer. |
| Bool | A Boolean value. | A single real value in $[0, 1]$, which can be interpreted as the probability that this Boolean value is true. |

Table 1: The type system of the domain-specific language for visual reasoning.

| Signature | Note |
|---|---|
| $scene()\longrightarrow$ ObjectSet | Return all objects in the scene. |
| $filter(\mathbf{a}$: ObjectSet, $c$: ObjConcept$)\longrightarrow$ ObjectSet | Filter out a set of objects having the object-level concept (e.g., red) from the input object set. |
| $relate(\mathbf{a}$: ObjectSet, $\mathbf{b}$: ObjectSet, $c$: RelConcept$)\longrightarrow$ ObjectSet | Filter out a set of objects in set $\mathbf{a}$ that have the relational concept (e.g., left) with the input object $\mathbf{b}$. |
| $intersection(\mathbf{a}$: ObjectSet, $\mathbf{b}$: ObjectSet$)\longrightarrow$ ObjectSet | Return the intersection of set $\mathbf{a}$ and set $\mathbf{b}$. |
| $union(\mathbf{a}$: ObjectSet, $\mathbf{b}$: ObjectSet$)\longrightarrow$ ObjectSet | Return the union of set $\mathbf{a}$ and set $\mathbf{b}$. |
| $query(\mathbf{a}$: ObjectSet, $c$: Attribute$)\longrightarrow$ ObjConcept | Query the attribute (e.g., color) of the input object $\mathbf{a}$. |
| $exist(\mathbf{a}$: ObjectSet$)\longrightarrow$ Bool | Check if the set is empty. |
| $count(\mathbf{a}$: ObjectSet$)\longrightarrow$ Integer | Count the number of objects in the input set. |

Table 2: All operations in the domain-specific language for visual reasoning.

then sent to a ResNet-34 [3] to extract a region-based representation (by RoI Align) and image-based representation, respectively. We concatenate them to form a vector embedding for each object in the image.

**Neuro-symbolic realization.** The high-level idea for the program execution is to build a collection of functions that realize the semantics of each operation based on the vector embeddings of objects and concepts. Taking the *filter* operation as an example, denote $\mathbf{a}$ as a vector representation of the input set, $o_i$ the object embeddings, and $e_c$ the concept embedding. We compute the vector representation $\mathbf{b}$ of the output set as:

$$\mathbf{b}_i = \mathbf{a_i} \cdot \sigma\left(\langle o_i, e_c \rangle\right),$$

where $\sigma$ is the sigmoid function, and $\langle \cdot, \cdot, \rangle$ is the inner product of two vectors. Intuitively, we first compute the inner product between the concept embedding $e_c$ and each object embedding, which gives as a vector of scores of whether object $i$ has concept $c$. Next, we compute the element-wise multiplication between two vectors.

| Program | Type | Value |
|---|---|---|
| *scene*() | ObjectSet | $[1, 1, 1]$ |
| *filter*(*scene*(), **CUBE**) | ObjectSet | $[0.8, 0.1, 0.9]$ |
| *filter*(*filter*(*scene*(), **CUBE**), **SHINY**) | ObjectSet | $[0.08, 0.08, 0.81]$ $= [0.8, 0.1, 0.9] \odot [0.1, 0.8, 0.9]$ |
| *count*(*filter*(*filter*(*scene*(), **CUBE**), **SHINY**)) | Integer | $0.97 = sum([0.08, 0.08, 0.81])$ |

Table 3: An illustrative execution trace of the program *count*(*filter*(*filter*(*scene*(), **CUBE**), **SHINY**)). *sum* denotes the "reduced sum" operation of a vector, which returns the summation of all entries in that vector. $\odot$ denotes element-wise multiplication for two vectors.

A key difference between our realization of these operations and the one in Mao et al. [8] is that we use element-wise multiplication to simulate the intersection between two sets, and $1 - (1 - \mathbf{a})(1 - \mathbf{b})$ for union. In contrast, Mao et al. [8] use element-wise min operation for intersection and max for union. Both realizations can be motivated by real-valued logic: product logic *vs*. Gödel logic. The main purpose of using products instead of min-max's is to make our realization compatible with our expected execution mechanism, which we will detail in Appendix 2.

**Example.** Here we run a concrete example to illustrate the execution process of a program in the visual reasoning domain. Suppose we have an image containing three objects $o_1$, $o_2$ and $o_3$. We have two additional vector embeddings for concepts **SHINY** and **CUBE**. Furthermore, $\sigma(\langle o_i, e_{\textbf{SHINY}} \rangle) = [0.1, 0.8, 0.9]$, and $\sigma(\langle o_i, e_{\textbf{CUBE}} \rangle) = [0.8, 0.1, 0.9]$.

Consider the input sentence "*How many shiny cubes are there*". Table 3 illustrates a step-by-step execution of the underlying program: *count*(*filter*(*filter*(*scene*(), **CUBE**), **SHINY**)).

**Expected execution.** In the visual reasoning domain, we have only implemented the expected execution mechanism for subordinate program trees whose type is *objset*, although many other types such as *integer* and *bool* also naturally supports expected execution. This is because, types such as *integer* and *bool* only appear at the sentence-level, and thus computing the "expectation" of such programs does not reduce the overall complexity.

Formally, the expected execution process compresses a list of semantic programs $v_1, v_2, \cdots, v_K$ and their corresponding weights $\tau(v_i)$ into a single semantic program $v^*$ with weight $\tau(v^*)$. Suppose all $v_i$'s have type *objset*. We use $\bar{v}_i$ to denote the execution result of these programs. Each of them is a vector of length $N$, where $N$ is the number of objects in the scene. We compute $\bar{v}^*$ and $\tau(v^*)$ as the following:

$$\bar{v}^* = \frac{1}{\sum_i \exp(\tau(v_i))} \sum_i \left( \exp(\tau(v_i)) \cdot \bar{v}_i \right),$$

$$\tau(v^*) = \log \sum_i \exp(\tau(v_i)).$$

Intuitively, we normalize the weights using a softmax function to translate them into a distribution. Then we compute the expectation of the vectors. For more details about the definition and properties of expected execution, please refer to our main text and Appendix 2.

**Candidate lexicons.** Recall that the process of lexicon learning has three stages. First, we generate an extensive collection of candidate semantic programs. Second, we generate candidate lexicon entries for each word by enumerating all possible candidate semantic programs generated in the first step and all possible ordering (linearization in a sentence) of its arguments. Third, we apply our CKY-E2 and gradient-based optimization to update the weights associated with each lexicon entry.

In our visual reasoning domain, we only consider the following candidate semantic programs and linearizations:

    1. Syntactic type: *objset*, semantic program: *scene*() (English noun).

2. Syntactic type: *objset*, semantic program: *filter*(*scene*(), **?**) (English noun).

3. Syntactic type: *objset/objset*, semantic program: $\lambda x.filter(x, ?)$ (English adjective).

4. Syntactic type: *objset\objset/objset*, semantic program: $\lambda x.\lambda y.relate(x, y, ?)$ (English preposition I).

5. Syntactic type: *objset\objset/objset*, semantic program: $\lambda x.\lambda y.relate(y, x, ?)$ (English preposition II)

6. Syntactic type: *bool/objset*, semantic program: $\lambda x.exist(x)$.

7. Syntactic type: *integer/objset*, $\lambda x.count(x)$.

8. Syntactic type: *word/objset*, $\lambda x.query(x, ?)$.

9. Syntactic type: CONJ$_{\text{AND}}$, $\lambda f.\lambda g.(\lambda x.intersect(f(x), g(x)))$ (generalized conjunction).

10. Syntactic type: CONJ$_{\text{OR}}$, $\lambda x.\lambda y.(\lambda z.intersect(z, union(x, y)))$ (generalized disjunction).

As we will see later, when we compare the candidate lexicon entries for the visual reasoning domain and the language-driven navigation domain, the visual reasoning domain contains significantly fewer entries than the navigation domain. This is because much of the learning process in this domain is associated with learning the concept embeddings. In the following few paragraphs, we will explain how we instantiate concepts based on these lexicon entry templates and implement generalized conjunction and disjunction in our domain.

First, for each word (more precisely, word type), e.g., *shiny*, we will instantiate 10 lexicon entries. For semantic programs that contain unbounded concept arguments (**?** marks), we will introduce a series word-type-specific concepts. Specifically in this domain, each word type will be associated with 3 concept representations: **SHINY**$_{\text{ObjConcept}}$, **SHINY**$_{\text{RelConcept}}$, and **SHINY**$_{\text{Attribute}}$. Based on Table 2, the first two concepts will be represented as two embedding vectors, and the the third concept will be represented as a vector, indicating which concepts belong to this attribute category. Next, we will instantiate these lexicon entries by filling in these concept representations. For example, one of the candidate lexicon entry for *shiny* is syntactic type: *objset*, semantic program: *filter*(*scene*(), **SHINY**$_{\text{ObjConcept}}$). During training, all these vector embeddings as well as the weights associated with each lexicon entry, will be optimized jointly.

Next, we discuss the implementation for two conjunctive lexicon entries. The grammar rule for CONJ$_{\text{AND}}$ is:

$$T \ \text{CONJ}_{\text{AND}} \ T \to T,$$

where $T$ is an arbitrary syntactic type (thus called generalized conjunction). There are two typical use cases: what is the shape of the red and shiny object, and what is the shape of the object that is left of the cube and right of the sphere. In the first case, both arguments have syntactic type *objset/objset*. In the second case, both arguments have syntactic type *objset\objset*. Note that CLEVR contains only the second case.

The grammar rule for CONJ$_{\text{OR}}$ is:

$$objset \ \text{CONJ}_{\text{OR}} \ objset \to objset\backslash objset.$$

It covers the case: *how many objects are blue cubes or red spheres.* Our implementation is slightly different with human-defined lexicon entries for the word *or*, in particular, because the DSL we use is a small set of set-theoretic operations, which does not fully match the expressiveness of truth-conditional semantics. Thus, the current DSL does not support the representation of all words in the dataset (in particular, *or* and *are*). Thus, we have implemented this ad-hoc fix to handle disjunction.

Finally, we want to emphasize again that, since our DSL does not support representing all semantic programs of words, we allow certain words to be associated with an "*empty*" lexicon entry. This entry can be combined with any words or constituents next to it and does not participate in the composition of syntactic types and semantic programs. In Table 4 we show the lexicon entry associated with each word in the sentence "*are there any shiny cubes?*", learned by our model, G2L2.

## 1.2 Language-Driven Navigation DSL

Our DSL for the language-driven navigation domain is a simple string manipulation language that supports creating new strings, concatenating two strings, and repeating a string multiple times. Our DSL contains only two primitive types: action sequence, abbreviated as ActSeq, and integer.

| Word Type | Syntactic Type | Semantic Program |
|-----------|----------------|------------------|
| are | <EMPTY> | <EMPTY> |
| there | <EMPTY> | <EMPTY> |
| any | $bool/objset$ | $\lambda x.exist(x)$ |
| shiny | $objset/objset$ | $\lambda x.filter(x, \textbf{SHINY}_{\text{ObjConcept}})$ |
| cubes | $objset$ | $filter(scene(), \textbf{CUBE}_{\text{ObjConcept}})$ |

Table 4: The learned lexicon entries associated with each word for a simple sentence: *are there any shiny cubes?*. The derived semantic program for the full sentence is $exist(filter(filter(scene(), \textbf{CUBE}_{\text{ObjConcept}}), \textbf{SHINY}_{\text{ObjConcept}}))$

| Signature | Note |
|-----------|------|
| $empty() \longrightarrow$ ActSeq | Create an empty string (of length 0). |
| $newprim() \longrightarrow$ ActSeq | Create a string containing only one primitive action. In SCAN, there are in total 6 primitives. |
| $newint() \longrightarrow$ Integer | Create a single integer. In SCAN, we only support integers $\{2, 3, 4\}$. |
| $concat(\mathbf{a}\text{: ActSeq}, c\text{: ActSeq}) \longrightarrow$ ActSeq | Concatenate two input strings. |
| $repeat(\mathbf{a}\text{: ActSeq}, \mathbf{b}\text{: Integer}) \longrightarrow$ ActSeq | Repeat the input string for multiple times. |

Table 5: All operations in the domain-specific language for language-driven navigation.

Formally, we summarize the list of operations in our language-driven navigation domain in Table 5.

**Probabilistic string representation.** We represent each string in a "probabilistic" manner. In particular, each string $s$ is represented as a tuple $\langle L^s, C^s \rangle$. $L^s$ is a categorical distribution of the length. $C^s$ is a three-dimensional tensor, indexed by $\ell, k, c$, where $C^s_{\ell,k,c} = p(s[k] = c|length(s) = \ell)$. Thus, $C$ has the shape $[L + 1, L, |V|]$, where $L$ is the max length of a string and $V$ is the action vocabulary. For simplicity, we constrain that $C^s_{\ell,k,c} \equiv 0$ for all $k > \ell$.

It is straightforward to represent empty strings: $L_0 = 1$, or strings with a single action primitive $a$: $L_1 = 1$ and $C_{1,0,a} = 1$. Now we explain our implementation of the *concat* and the *repeat* operation.

For $z = concat(x, y)$:

$$L^z_\ell = \sum_{0 \le i \le \ell} \left( L^x_i \cdot L^y_{(\ell-i)} \right);$$

$$C^z_{\ell,k,c} = \frac{1}{L^z_\ell} \sum_{0 \le i \le \ell} \left( L^x_i \cdot L^y_{(\ell-i)} \cdot (C^x_{i,k,c} + C^y_{\ell-i,k-i,c}) \right).$$

The high-level idea is to enumerate the possible length of both strings.

Similarly, for $z = repeat(x, m)$,

$$L^z_\ell = \begin{cases} L^x_{\ell/m} & \text{if } \ell \bmod m = 0 \\ 0 & \text{otherwise} \end{cases}$$

$$C^z_{\ell,k,c} = \begin{cases} L^x_{\ell/m, k \bmod (\ell/m), c} & \text{if } \ell \mod m = 0 \text{ and } k < \ell \\ 0 & \text{otherwise} \end{cases}.$$

**Expected execution.** In the language-driven navigation domain, we only perform expected execution for semantic programs of type ActSeq, whose execution results can be represented using the probabilistic string representation. Denote $\bar{s}_i$ as the execution results for $K$ programs, and $\tau(s_i)$ the corresponding weights. We define $p(s_i) = softmax\left(\{\tau(s_i)\}\right)_i = \frac{\exp \tau(s_i)}{\sum_j \exp \tau(s)j}$. We compute the

| Type | Program (Note) |
|---|---|
| ActSeq | $walk()$ 
 The simplest program that constructs a string with a single action primitive: **WALK**. |
| (ActSeq) $\longrightarrow$ ActSeq | $\lambda x.concat(look(), x)$ 
 Prepend a **LOOK** action to an input string. |
| (ActSeq, ActSeq) $\longrightarrow$ ActSeq | $\lambda x.\lambda y.concat(x, y)$ 
 Concatenate two strings. |
| (ActSeq, ActSeq) $\longrightarrow$ ActSeq | $\lambda x.\lambda y.concat(repeat(x, 2), y)$ 
 Repeat the first string twice and concatenate with the second string. |
| ((ActSeq) -> ActSeq, ActSeq) $\longrightarrow$ ActSeq | $\lambda x.\lambda y.concat(y, x(walk()))$ 
 The first argument $(x)$ is a function which maps a ActSeq to another ActSeq. The second argument $y$ is an ActSeq. The function invokes $x$ with a simple string **WALK**, and concatenate the result with $y$. |

Table 6: Sample semantic programs generated by the enumeration process based on our language-driven navigation DSL.

expected string $\bar{s}$ and its weight $\tau(s)$ as:

$$
L_\ell^s = \sum_i p(s_i) L_\ell^{s_i}; \quad C_{\ell,k,c}^s = \frac{\sum_i \left( p(s_i) L_\ell^{s_i} \cdot C_{\ell,k,c}^{s_i} \right)}{L_\ell^s}.
$$

**Candidate lexicons.** We use a simple enumerate algorithm to generate candidate lexicon entries for our language-driven navigation DSL. Specifically, we first enumerate candidate semantic programs for each lexicon entry that satisfy the following constraints:

1. There are at most three operations.
2. There are at most two arguments.
3. There is at most one argument whose type is a functor.
4. There is no argument of type *Integer*.

Table 6 lists a couple of programs generated by the algorithm and their corresponding types.

Based on the candidate semantic types, we first instantiate candidate lexicon entries by enumerate possible ordering (linearization) of the arguments. For example, the simple program $\lambda x.concat(look(), x)$ has two possible linearizations: *ActSeq/ActSeq* and *ActSeq\ActSeq*. As discussed in the main paper, in order to handle parsing ambiguities, we further introduce two finer-grained syntactic types for the *ActSeq* type: *S* and *V*. In practice, we only allow the following set of syntactic types: *V*, *V/V*, *V\V*, *V\V/V*, *V\V/(V\V)*, and *S\V/V*. In total, we have 178 candidate lexicon entries for each word.

| Word Type | Syntactic Type | Semantic Program | exp(Weight) |
|---|---|---|---|
| ONE | N | 1 | 1.0 |
| PLUS_ONE | N\N | $\lambda x.x + 1$ | 0.5 |
| PLUS_ONE | N\N | $\lambda x.x \times 3$ | 0.5 |
| MUL_THREE | N\N | $\lambda x.x + 1$ | 0.5 |
| MUL_THREE | N\N | $\lambda x.x \times 3$ | 0.5 |

Table 7: A set of candidate lexicon entries and their weights in a simple arithmetic domain.

| Index | Syn. Type | Semantic Program ($derivation_k$) | Execution Result ($exec_k$) | exp(Weight) |
|---|---|---|---|---|
| 1 | N | $(1 + 1) + 1$ | 3 | $0.25 = 1 \times 0.5 \times 0.5$ |
| 2 | N | $(1 \times 3) + 1$ | 4 | $0.25 = 1 \times 0.5 \times 0.5$ |
| 3 | N | $(1 + 1) \times 3$ | 6 | $0.25 = 1 \times 0.5 \times 0.5$ |
| 4 | N | $(1 \times 3) \times 3$ | 9 | $0.25 = 1 \times 0.5 \times 0.5$ |

Table 8: Four candidate derivations of the simple sentence "*ONE PLUS_ONE MUL_THREE*" in a simple arithmetic domain.

## 2 Delve Into Expected Execution

In this section, we will run a concrete example in a small arithmetic domain to demonstrate the idea of expected execution. Following that, we will prove an important invariance property that has guided our realization of different functional modules in both domains.

### 2.1 CKY-E2 In An Arithmetic Domain

In this section, we will consider parsing a very simple sentence in an arithmetic domain. We will be using numbers and two arithmetic operations: $+$ and $\times$. Each number in the domain will be represented as a real value.

Suppose we have the following lexicon entries associated with three words, illustrated in Table 7. There are 4 candidate derivations of the sentence "*ONE PLUS_ONE MUL_THREE*", as illustrated in Table 8. For simplicity, we will show the $\exp$ of weights. Thus, the probability of a derivation is proportional to the product of all lexicon entry weights.

Suppose that we will be using the groundtruth execution result of this program as the supervision, applied by an L2 loss. Then we will be interested in the expected execution result of all possible derivations. In this case, it is

$$\mathbb{E}[exec] = 0.25 \times 3 + 0.25 \times 4 + 0.25 \times 6 + 0.25 \times 9 = 5.5.$$

Next, we will try to accelerate the computation of $\mathbb{E}[exec]$ by doing local marginalization. Consider the constituent "*ONE PLUS_ONE*". In CKY-E2, this will be the first constituent that the algorithm constructs. It has two possible derivations, whose corresponding semantic programs are $(1 + 1)$ and $(1 \times 3)$. Both derivations have the same syntactic type N, and thus, they will be combined with N\N on its right, in the next step. In this case, in CKY-E2, we will merge these two derivations into one (again, since we only care about the expected execution result, not the set of all possible derivations!). The combined derivation has value $0.5 \times (1 + 1) + 0.5 \times (1 \times 3) = 2.5$, and total weight $0.5 + 0.5 = 1$.

Then, when we are trying to compose the derivation for the whole sentence, i.e., combine the constituent "*ONE PLUS_ONE*" and "*MUL_THREE*", we no longer need to compute all $2 \times 2 = 4$ possible derivations, but only $1 \times 2 = 2$ derivations. They are: $2.5 + 1 = 3.5$, with probability 0.5, and $2.5 \times 3 = 7.5$, with probability 0.5. In this case, we see that taking local marginalization reduces the computation complexity of parsing and retains the expected execution result!

## 2.2 Formal Properties of CKY-E2

Motivated by the intuitive example shown above, now let us formally specify the properties of CKY-E2.

**Expectation invariance.** Consider the composition of two consecutive constituents $a$ and $b$. We use $a_1, \cdots a_N$ and $b_1, \cdots b_N$ to denote possible derivations of both constituents. We assume all $a_i$'s are of the same syntactic type without loss of generality, so do all $b_i$'s, since we will handle different syntactic types separately.

Denote $f$ as the semantic composition function for $a$ and $b$. Without local marginalization, we will have in total $N \times M$ derivations for the result constituent: $c_{i,j} = f(a_i, b_j)$. We further use $\bar{a}_i, \bar{b}_j$, and $\bar{c}_{i,j}$ to denote the execution results of these derivations. Without derivations, the expected execution results is:

$$
\begin{aligned}
\mathbb{E}[\bar{c}] &= \frac{1}{\sum_{i,j} \exp \tau(c_{i,j})} \sum_{i,j} \left( \exp \tau(c_{i,j}) \cdot \bar{c}_{i,j} \right) \\
&= \frac{1}{\sum_{i,j} \exp \tau(a_i) \cdot \exp \tau(b_j)} \sum_{i,j} \left( \exp \tau(a_i) \cdot \exp \tau(b_j) \cdot f(\bar{a}_i, \bar{b}_j) \right).
\end{aligned}
$$

Again, without loss of generality, we will assume $\sum_i \tau(a_i) = 1$ and $\sum_j \tau(b_j) = 1$, because any constant scaling of these weights will not change the expectation $\mathbb{E}(\bar{c})$. Thus, we simplify this definition as,

$$
\mathbb{E}[\bar{c}] = \sum_{i,j} \left( \exp \tau(a_i) \cdot \exp \tau(b_j) \cdot f(\bar{a}_i, \bar{b}_j) \right).
$$

Let us assume function $f$ has the following property: $\mathbb{E}[f(a,b)] = f(\mathbb{E}(a), \mathbb{E}(b))$, which expands as,

$$
\sum_{i,j} \left( \exp \tau(a_i) \cdot \exp \tau(b_j) f(\bar{a}_i, \bar{b}_j) \right) = f \left( \sum_i \left( \exp \tau(a_i) \bar{a}_i \right), \sum_j \left( \exp \tau(b_j) \bar{b}_j \right) \right).
$$

Thus, locally marginalizing the expected value for $\mathbb{E}[\bar{a}]$ and $\mathbb{E}[\bar{b}]$ will not change the expected execution result of $c$.

In this simple proof we have made a strong assumption on the composition function $f$: $\mathbb{E}[f(a,b)] = f(\mathbb{E}(a), \mathbb{E}(b))$. In practice, this is true when $f$ are addition or multiplication functions between scalars, vectors, matrices, and in general, tensors. It will not apply to element-wise min/max operations and other non-linear transformations. Fortunately, this already covers most of the operations we use in the visual reasoning and language-driven navigation DSLs. In practice, even if some operations do not have this property, we can still use this mechanism to approximate the expected execution result.

Although we have only proved this property for binary functions, the idea itself easily generalizes to unary functions, such as the negation operation, and higher-arity functions. Furthermore, by induction over derivation trees, we can easily prove that, as long as all composition functions satisfy the expectation invariance property, applying CKY-E2 yields the same result as doing the marginalization at a sentence level.

**Complexity.** In general, it is hard to quantify the reduction in computational complexity by doing local marginalization. However, we can still estimate the number of possible derivations constructed in the entire CKY-E2 procedure. For simplicity, consider the case where there is only one primitive syntactic type: *X*. Moreover, there are $N_0$ candidate lexicon entries of type *X*; $N_1$ entries of type *X/X*, and $N_2$ entries of type *X\X/X*. For each span, considered in the CKY-E2 algorithm, all possible derivations associated with this span can be grouped into 4 categories:

1. Derivations of type *X*. In this case, only 1 derivation will be retained (merged by the expected execution result).

2. Derivations of type *X/X*. In this case, they must be a primitive lexicon entry. Thus, there are at most $N_1$ of them.

3. Derivations of type *X\X/X*. Similarly, at most $N_2$ of them.

4. Derivations of type *X\X*. This intermediate syntactic type is a result of a partial composition between *X\X/X* and *X* (on its right). Thus, there are at most $N_2$ of them.

Overall, there are at most $1 + N_1 + 2 \times N_2$ derivations stored for this span. Since the total span is $O(L^2)$, where $L$ is the total length of the input sentence, the overall complexity of CKY-E2 is a polynomial of $L$, $N_0$, $N_1$, and $N_2$, which is significantly lower than an exponential number of derivations.

## 2.3 Connection with Other Parsing Models

**Connection with synchronous grammars.** Our approach can be viewed as defining a synchronous grammar over joint (parse tree, meaning program) pairings. We didn't use this framing in the main paper because typical applications of synchronous grammar involve parallel datasets (e.g., sentence pairs in two languages for machine translation, or sentence–image pairs for generating image descriptions) in which the information in both modalities is directly parsed. In our setting, in contrast, the meaning-program component of the synchronous grammar is acquired through more distant supervision. We will make all this clear in the final version, including stating how the chart parsing process can be seen as synchronously constructing parsing trees and meaning programs. The expected execution can be viewed as a "compression" step over all meaning programs that can be potentially parsed from each span.

**Connection with sum-product CKY.** There are also connections between CKY-E2 and sum-product CKY, such as the shared Markovian assumption, but would like to add that the main difference between CKY-E2 and sum-product CKY is that CKY-E2 computes only the "expectation" of the execution results of the underlying program. Instead, sum-product CKY computes a full distribution of the parsing results (e.g., in syntax parsing, it can compute the categorical distribution of the root symbol). Sum-Product CKY can not be applied to our setting, because we are dealing with programs and the space of possible programs is infinite. Modeling a distribution of all possible programs might be intractable; instead, computing the expectation is much easier, and this enables us to do local marginalization.

# 3 Experimental Setup And Analysis

In this section, we will present in detail the experimental setups for both datasets: CLEVR and SCAN. Both datasets are released under a BSD license. Specifically, we will include details about the setups for different compositional generalization tests. Although some of them (the ones in the SCAN dataset) have already been illustrated in their original paper, we echo them here for completeness. Next, we will also analyze the performance of each model on both datasets, focusing on the inductive biases they have in their model and how these inductive biases contribute to their compositional generalization in different splits.

## 3.1 Visual Reasoning: CLEVR

We start with the dataset generation process for the CLEVR dataset. Next, we formally present the dataset generation protocol for all splits. Furthermore, we analyze the performance of various models.

**Baselines.** As a quick recap of our baselines, MAC [5] uses an end-to-end vision-language attention mechanism to process the question and image jointly; TbD-Nets [9] and NS-VQA [11] uses a neural sequence-to-sequence model (with attention, see [2] for semantic parsing. The parser is trained with sentence-program pairs; NS-CL [8] uses a customized sequence-to-tree model, and jointly learns the visual recognition models and the semantic parser. One crucial implementation detail with the semantic parser module in NS-CL is that, it uses additional token embeddings to annotate the concepts appearing in the question. When generating a concept token in the semantic program, it uses an attention mechanism to select the concept from the input question.

**Dataset generation.** Since we only consider the cases where each word is associated with a unique lexicon entry, we have manually excluded sentences that will break this assumption. Among all of the 425 templates in the original CLEVR dataset [6], we have retained 208 templates. Specifically, we have removed all templates that involve: 1) coreference resolution, 2) "same"-related questions, and 3) number comparison-related questions. To keep the number of questions the same as the original dataset, we choose to re-generate the questions using the selected subset of templates, following the original data generation protocol. All our splits are generated based on this basic version, which we name as the standard training set and the standard test set.

**Split: data efficiency.** We test the data efficiency of models by only using 10% of training data in the standard training set, and test the models on the standard test set.

In this split, the semantic parsers used by all program-based methods: TbD-Nets, NS-VQA, and NS-CL, have nearly perfect accuracy. Thus, the performance drops are primarily due to the limited data for training individual modules. Overall, TbD-Nets have the worst data efficiency. There is no performance drop for the NS-VQA model, because the visual recognition modules are pretrained with direct object-level supervision.

**Split: compositional generalization (purple).** The training set is generated by selecting all questions that either do not contain the word "purple" or have a length smaller than or equal to 7 (including punctuation). The test set is generated by selecting all sentences containing the word "purple" and has a length greater than 7.

In this split, the semantic parsers used by all program-based methods: TbD-Nets, NS-VQA, and NS-CL have nearly perfect accuracy. Thus, the performance drops are primarily due to 1) the limited data for training individual modules (in this case, *filter*(*purple*) and 2) novel composition of learned modules. Overall, NS-VQA and NS-CL answer more questions correctly than TbD-Nets.

**Split: compositional generalization (right).** The training set is generated by selecting all questions that either do not contain the word "right" or have a length smaller than or equal to 12. The test set is generated by selecting all sentences that contain the word "right" and have a length greater than 12.

In this split, the semantic parser of NS-CL still yields almost perfect accuracy. In contrast, the accuracy of the neural sequence-to-sequence parser used by TbD-Nets and NS-VQA is around 91%. Thus, the performance drop of TbD-Nets is mainly due to the inferior performance of the

corresponding neural module: *relate*(*right*). Compared with the realization of a *filter* module (used in our *purple* generalization test), the *relate* module in TbD-Nets has a significantly deeper neural architecture (6 layers vs. 3 layer). Thus, we hypothesize that this module requires more data to train.

**Split: compositional generalization (count).** The training set is generated by selecting all questions that either do not contain operation "count" or have a length smaller than or equal to 9. The test set is generated by selecting all sentences that contain operation "count" and have a length greater than 9.

Among all compositional generalization tests, this is the most challenging one. The semantic parser, in this case, need to generalize from: "*how many cubes are there*" and "*what's the shape of the object that is both left of the cube and right of the sphere?*", to "*how many cubes are both left of the cube and right of the sphere?*" We have constructed the training data in a way such that all constituents have been seen in the training data. In this test, the program-level accuracy of the semantic parser used by TbD-Nets and NS-VQA is 70.8%. NS-VQA outputs slightly higher QA accuracy.

We find that, for the parsing module in NS-CL, it fails to output the correct filter operation. Given the input question "what is the number of spheres that are right of the cube", it sometimes outputs *filter_cube relate_right filter_cube count* (the correct program has the third operation *filter_sphere* instead). This is because the system has never seen sentences composed of "counting" operations and such complex structures; during training, it has only seen short sentences such as "what is the number of spheres?" Only our G2L2 model, with its explicit constituent-level compositionality, is able to make these generalizations.

**Split: depth generalization.** We define the "hop number" of a question as the number of intermediate objects being referred to in order to locate the target object. For example, the "hop number" of the question "*how many red objects are right of the cube?*" is 1. We train different models on 0-hop and 1-hop questions and test them on 2-hop questions.

This generalization test evaluates the generalization to deeper syntactic structures. All methods except for our model G2L2 fail on this test. By evaluating the accuracy of the program generated by different semantic parsers, we find that, the neural sequence-to-sequence model used by TbD-Nets and NS-VQA completely fails on this task, sometimes generating invalid programs (the program-level accuracy is 1.7%). Thus, we see a significant performance drop for both methods. Meanwhile, NS-CL also generates wrong programs, but the programs are always valid due to its sequence-to-tree design. Furthermore, even if the program is not correct, for example, they miss certain operations, the execution result may still lead to a correct answer. For example, as long as the semantic parser gets the outer-most filter operation (i.e., the last hop) correct, it is still possible to generate the correct answer.

## 3.2 Language-Driven Navigation: SCAN

The SCAN dataset [7] consists of several official splits for generalizability evaluation. Following existing work, we evaluate on the splits corresponding to the generalization test of "jump" and "around right". In addition, we test the generalizability across different output lengths and the data efficiency of models. The performance is measured by exact match–based output accuracy.

**Split: data efficiency.** The official "simple" split randomly samples 80% among all possible example pairs as the training set, and leaves the others as the test set. We use the training set of the official simple split (available at [this url]) as the entire training set, and test the data efficiency by using only 10% of them. We sample the 10% data uniformly for each input length. Both settings are tested in the official simple test split, which is available at [this url].

**Split: compositional generalization (jump).** The training split consists of *jump* in isolation, i.e., the input is *jump* while the ground-truth output is I_JUMP, along with all other examples that do not contain *jump*. The model is expected to recognize that *jump* has the same syntactic category as other verbs such as *run*, and does well on complicated instructions including *jump*, e.g., mapping *jump twice* to I_JUMP I_JUMP. The training data is available at [this url] and the test data is available at [this url].

**Split: compositional generalization (around right).** Similar to the "jump" test, the training set for the "around-right" test consists of all possible examples that do not contain *around right* in their inputs, while the test set consists of all examples that have *around right*. It is worth noting that different from *jump*, *around right* in isolation is not a valid input as it lacks a primitive. The model is expected to perform compositional generalization, understanding *around right* based on existing training phrases such as *around left* and *opposite right*. The training data is available at [this url] and the test data is available at [this url].

**Split: length generalization.** The model is expected to perform well on examples with long ground-truth output while training those with short ground-truth output. In this test, all training examples consist of less than or equal to 22 tokens in their outputs, while the output of a test example may consist of up to 48 tokens. The training data is available at [this url] and the test data is available at [this url].

**Baseline models.** All baseline models are built on top of a seq2seq model [10]: the original seq2seq model [10] trains an LSTM-based encoder-decoder model using the training set; the other methods augment the training set by either heuristics or learned models, and train an LSTM-based encoder-decoder model using the augmented data. It is worth noting that among all considered baseline methods, GECA [1] may generate examples in the test set, especially for compositional generalization tests since the heuristics it introduces is by nature compositional.