# OpenReview forum: "Grammar-Based Grounded Lexicon Learning"
_NeurIPS.cc/2021/Conference — NeurIPS 2021 Poster_

### Official Review · Reviewer_45Px · 2021-07-01

**Rating:** 5
**Confidence:** 3

**Summary:**

This paper introduces a new method, G^2L^2 (Grammar-Based Grounded Lexicon Learning).

The task is CLEVR: https://cs.stanford.edu/people/jcjohns/clevr/ (and SCAN)

For CLEVR, inputs are pictures + questions such as:

Q: Are there an equal number of large things and metal spheres?

Q: What size is the cylinder that is left of the brown metal thing that is left of the big sphere?

Q: There is a sphere with the same size as the metal cube; is it made of the same material as the small red sphere?

Q: How many objects are either small cylinders or red things?

Outputs are simple answers that can be scored with simple mechanisms such as string matching.

According to Table 1, the proposed method is about as good as some others in the literature (NS-VQA, NS-CL) on the standard task, but if we restrict ourselves to some interesting subcases such as the last two columns, the proposed method is considerably better.

The proposed method starts with NS-CL, but adds the lexicalist hypothesis, a linguistic framework that was popular a few decades ago (the HPSG reference is from 1994, and the Bresnan reference is even more recent, but Wikipedia traces the history back to the 1950s https://en.wikipedia.org/wiki/Lexicalist_hypothesis#History).   I suspect there is even more history than that, but nevertheless, for this audience, it may be more helpful to provide more tutorial background.  What is the lexical hypothesis, and how does it relate to this task?



**Main Review:**

This paper is hard to read.  It expects that the reader is familiar with quite a bit of difficult background.  The paper does not say much about CLEVR, or the baseline methods, or the main contribution (how the lexicalist hypothesis can improve over the baselines, at least in some interesting special cases such as the last two columns of Table 1).

I think this paper would have more impact if it was more accessible to a larger audience.  It is unlikely that people that know about vision and deep nets will also know about the lexicalist hypothesis, and vice versa.  Can this paper be written in a way that could work with an audience that knows about some of this background, but not not all of it?

I happen to be reasonably well-informed on much of the background, though I did spend some time chasing down references for some of these topics.  It is nice to see a connection between all these different interdisciplinary topics, but I would like to see more discussion on why the proposed method does as well as it does on the last two columns of Table 1?

If I wanted to be nasty, I could suggest that there are so many ways to split the data that one could always find a few that would make almost any solution look good.  I suspect that count and depth are fundamentally different from purple and right-of.  Suppose we have k objects, and each one has just a single color.  Then we can address simple properties like purple (color), but treating each object one-at a time.  Relations like right-of are slightly more complicated.  That might involve looking at pairs of objects.  Thus the search space is perhaps choose(k,1) for colors, and choose(k,2) for properties of pairs of objects like right-of.  I suspect the search space for counting is fundamentally more difficult, especially if we need to understand the implicit argument which probably involves some notion of equivalence relations.

Depth seems to be even more abstract.  I have to admit that I have read L252-256 a few times, but I still do not understand the definition of "hop number".

I suspect the main point is not so much about the lexicalist hypothesis, but more a statement about standard benchmarks such as CLEVR and SCAN.  It appears that there are certain aspects of these tasks that make them too-easy or too-hard for currently popular methods.  The field would benefit from a deeper understanding of these benchmarks.  Can we identify aspects of the tasks that popular methods are doing too well on, or where there are opportunities for improvement?  In some cases, simple patches such as rules may improve results by quite a bit.  When that happens,  what should we conclude?  Should we ignore the result because rules are out of fashion?  Or should we take it as a challenge to find a way to generalize popular methods to do better on subcases such as count questions and 2+ hop questions?

How realistic are these benchmarks?  Should we care about count questions?  2+ hop questions?  I thought the SCAN benchmark was introduced to make a theoretical point about the ability of certain methods to address certain questions involving compositionality.   Now that many methods appear to have solved that task, should we move on?

Perhaps the point is more about representation than the lexicalist hypothesis.  It makes sense to model words as predicates and arguments.  Categorical grammar can be viewed as a generalization of this.  These days, there is more interest in end-to-end training, and less interest in representation, but I could believe there are lots of opportunities for improvement involving representation.  But given the current popularity of end-to-end systems, the argument for representation needs to be much stronger than what was expected when there was more interest in representation, and less interest in end-to-end methods.


typo:
Caption table 1: Our model achives a comparable results


**Time Spent Reviewing:**

2

---

> ### Author Response · Authors · 2021-08-10
> **Author Response to Reviewer 45Px**
>
> Thank you for your helpful comments.
>
> ---
>
> **Q1**: More information about the lexicalist hypothesis and theories.
>
> **A1**: We’re delighted at your interest and will be glad to include more in our revision about lexicalism and its role in the bigger picture of developing neuro-symbolic methods for language. Narrowly, the “lexicalist hypothesis” whose Wikipedia page you refer to is a specific theory that the morphological processes of word formation are different from the syntactic processes by which words combine into phrases and phrases recursively combine with each other. More broadly, the lexicalist theories of syntax (which grew out of the lexicalist hypothesis) that are of interest to us propose that (i) the key syntactic principles by which words and phrases combine are extremely simple and general, and (ii) nearly all of the complexity in syntax can be attributed to rich and detailed lexical entries for the words in the language. For example, whereas the relationship between the active and passive voice, e.g. “Kim saw a balloon” versus “A balloon was seen by Kim”, was treated in pre-lexicalist theories as a special syntactic rule converting between the sentences, in lexicalist theories this relationship derives simply from the knowledge that the passive participle for the verb “see” is “seen”, which interacts with knowledge of other words to make both the active and passive forms of the sentence possible. In lexicalist theories, the problem for the language learner thus becomes a problem of learning the words in the language, not a problem of learning numerous abstract rule schemas. The Combinatory Categorial Grammar (CCG) framework we use is a well-established example of a lexicalist theory: there is a universal inventory of just three combinatory rules (our Figure 1a), but those rules can only be applied once richly specified lexical entries are learned for the words in a sentence. We believe that this lexicalist-theory approach is a particularly good fit to the problem of grounded language learning: the visual context provides clues to the word’s meaning, and the word’s grammatical behavior is tied closely to this meaning, making learning efficient.
>
> ---
>
> **Q2**: CLEVR dataset, baselines, and results.
>
> **A2**: Due to the space limit, we have included the domain specific language we use in Supplementary Sec 1.1. We discuss the dataset setup, baseline analysis and the interpretation of the results in Sec 3.1, including the failure cases of baselines. We summarize the results related to the lexicalist hypothesis below. Please also see our general response.
>
> Overall, all baselines do not perform well in compositional generalization, particularly on challenging tasks such as counting and depth. MAC is based on multi-modal attention. TbD and NS-VQA uses attention-based sequence to sequence model for semantic parsing. NS-CL uses an attention-based sequence-to-tree model. These models generalize decently to scenarios where a single word is replaced. For example, in the “purple” generalization test, from “what’s the shape of the red object” to “what’s the shape of the purple object”. In both “counting” and “depth” generalization, the models need to generalize to sentences with long constituent-level recomposition, or even deeper structures. The lexicalist assumption is mostly helpful in these cases.
>
> ---
>
> **Q3**: Difference between “purple”, “right-of” and “count”.
>
> **A3**: The difference lies in two aspects: the interpretation of language, and the grounding of concepts. As described in A1, in the “purple” split, the program structures in the training set and the generalization test set are similar: they only differ in one operation.
>
> However, in the “counting” split, the problem is generalizing from “is there any spheres that are right of the cube” to “what is the number of spheres that are right of the cube”. We found that, for example, for the parsing module in NS-CL, it fails to output the correct `filter` operation. It sometimes outputs `filter_cube relate_right filter_cube count` (the correct program has the third operation `filter_sphere` instead). This is because the system has never seen sentences composed of “counting” operations and such complex structures; during training, it has only seen short sentences such as “what is the number of spheres?” Only our G2L2 model, with its explicit constituent-level compositionality, is able to make these generalizations.
>
> The depth case is even more difficult, where the system needs to generalize to sentences with deeper nested prepositional structures: from “what is the color of the cube left to the sphere?” to “what is the color of the cube left of the sphere that is behind the large metal thing?” Again, all models except G2L2 fail in these cases.
>
> The second difference lies in the hardness of concept learning, or concept grounding. The reviewer is right that questions with “count” are harder than “filter” because the hypothesis space is larger. We find that systems based on neural module networks (TbD-Nets) do not generalize to longer program sequences compared with neuro-symbolic reasoning models (e.g., NS-VQA, NS-CL, and G2L2). For example, compare the performance of TbD-Net and NS-VQA on “right of”. They use the same semantic parser but the performance of TbD-Net is far worse than NS-VQA.
>
> ---
>
> **Q4**: Hop number.
>
> **A4**: The “hop number” is the number of intermediate objects being referred to, or, the level of nested prepositions. “What is the color of the sphere” has hop number 1. “What is the color of the sphere left of the cube” has hop number 2. “What is the color of the sphere left of the sphere that is behind the large metal thing” has hop number 3.  We understand this is explained too briefly in the current draft, and will explain it more clearly in the revision.
>
> ---
>
> **Q5**: Difficulty of the benchmarks. What are the aspects of the tasks that popular methods are doing too well on, or where there are opportunities for improvement?
>
> **A5**: In the paper, we are trying to disentangle the grounded language learning problem into three parts: compositional understanding of language in the form of neuro-symbolic programs (parsing), compositional execution of the programs (reasoning), and learning the association between concepts and visual appearance (visual perception). Our results show that we have good tools for solving these challenges in specific datasets, but there are still many future directions for further improving these methods -- and that will require new and better benchmarks which our results can help point the way towards. In particular:
>
> For all methods, the visual perception part is typically handled using convolutional neural networks. In domains such as CLEVR, with a sufficient amount of data, all methods perform very well in this regard. However, how to learn new concepts with fewer examples as humans do, in a way that scales to a larger vocabulary of concepts, remains an open challenge.
> The reasoning part can be handled with either neuro-symbolic approaches or end-to-end approaches. We find that neuro-symbolic approaches (NS-VQA, NS-CL, and G2L2) have better data efficiency and stronger compositional generalization than methods that do not perform explicit reasoning over objects and their associated concepts (TbD and MAC, see our analysis in A3 above, and also our general response). However, current neuro-symbolic approaches still rely on certain domain-specific languages that only capture a limited portion of meanings that can be expressed in human language. Future work should consider more general representations of natural language semantics, and build datasets and metrics that test that.
> The compositional language understanding part has not been the focus for most papers targeting CLEVR. In CLEVR, we developed new train/test splits to test out-of-sample generalization to sentences with more complex compositional structure and greater embedding depth.  Across each of these tests, some baseline models performed well and others poorly, but our model was the only one that solved all these generalization tasks. However, we are still probing only a relatively small subspace of the compositionality that natural language offers and exploits. Future work very much needs new datasets and benchmarks that will test models’ broad ability to generalize compositionally. In particular, scaling up to handle sense-ambiguity is crucial: each word can have multiple meanings, active in different contexts, and these meanings can also have different grammatical properties. (See also A6 below.)
>
> In our revision, we cannot go into great detail on these points but we will include brief remarks about these important directions for future work in our discussion.

---

> ### Author Response · Authors · 2021-08-10
> **Author Response to Reviewer 45Px (cont.)**
>
>
> **Q6**: How realistic are these benchmarks? If we have solved them, should we move on?
>
> **A6**: We agree with the reviewer that these benchmarks are primarily created for testing compositional generalizations in certain aspects: language understanding and visual reasoning, more in a theoretical sense. But the compositional generalization tests we evaluate on in the paper, especially the hard ones that involve constituent-level (beyond simply word-level) composition of meaning, are important properties for models to have, for natural language understanding. G2L2 presents a candidate approach, which is the explicit incorporation of lexical grammar-based models for tackling these challenges. It is the only approach that solves all generalization tests.
>
> We see many meaningful future directions. Here we highlight one of them that we think is most important. The inductive biases of all models, including the data augmentation-based ones (e.g., GECA for SCAN) and grammar-based ones (G2L2) are primarily based on the idea that “word meanings are consistent across contexts” and that “sentence meaning is composed of word meaning.” We believe the next step is to extend this idea to more complex natural language, maybe in a specific domain, but with principled ways of analyzing the relationship between context-aware meaning of words and their consistency across contexts. On the context-aware extreme, tabular-style learning (e.g., memorizing all sentence-program mappings in the training set) will not generalize compositionally. On the context-free extreme, models can not handle sense variations and may fail in scaling to more free-form languages. Building benchmarks and design models that systematically handle this tradeoff is a meaningful direction.
>
> ---
>
> **Q7**: Should we put emphasis on representation, given that end-to-end methods are more popular?
>
> **A7**: We believe that the representation problem is important, even for end-to-end methods. For example, the end-to-end learning method MAC [3] is designed to emulate visual program execution. Recently there has also been interest in understanding the true representational power of neural networks that perform well in many language and visual reasoning tasks, such as graph neural networks and transformers [4,5]. Specifically in the visual reasoning domain, understanding how different models process queries and how they generalize are important questions. In this paper, we are seeing stronger generalization power by incorporating explicit program structures and disentangled concept learning and reasoning. On the language side, our model’s implementation of the lexicalist hypothesis improves the generalization of semantic parsing.  We agree that further work is needed to understand the nature of the representational advantages that come from combining a grammar-based lexical representation for language with grounded neuro-symbolic representations for visual reasoning, and how this compares with the representational power of pure end-to-end methods.
>
>
> [1] Jiayuan Mao, Chuang Gan, Pushmeet Kohli, Joshua B. Tenenbaum, and Jiajun Wu. "The neuro-symbolic concept learner: Interpreting scenes, words, and sentences from natural supervision." In ICLR 2019.
>
> [2] Jacob Andreas. “Good-Enough Compositional Data Augmentation.” In ACL 2020.
>
> [3] Drew A. Hudson, Christopher D. Manning. “Compositional Attention Networks for Machine Reasoning.” In ICLR 2018.
>
> [4] Pablo Barceló, Egor V. Kostylev, Mikael Monet, Jorge Pérez, Juan Reutter, Juan Pablo Silva. “The Logical Expressiveness of Graph Neural Networks.” In ICLR 2020.
>
> [5] Gail Weiss, Yoav Goldberg, Eran Yahav. “Thinking Like Transformers.” In ICML 2021.

---

> ### Author Response · Authors · 2021-08-27
> **Paper Revision**
>
> Dear Reviewer 45Px,
>
> Thanks for your time and consideration. We deeply appreciate the points you raised about adding more background material, a more thorough and conceptual presentation of our datasets and results, and more discussion of representation and future directions. We are already incorporating these points in our revision for the final version of the paper (although NeurIPS does not allow this to be shared with reviewers during the review process).
>
> In particular, our revised paper will include several additional paragraphs on the background of the lexicalist hypothesis for representing and learning natural language grammars, and a tutorial example of how this works for several simple sentences that are representative of those in our bigger datasets.  Then when we present our datasets and results, we will highlight how they embody the same phenomena from the tutorial introduction, and allow us to empirically assess the ability of different representations for learning these aspects of meaning in a scalable way.  These improvements are directly in response to your suggestions and will make the paper much stronger and more impactful, as you say.  We look forward to sharing our next version here, after the review period has ended, and we are sorry that NeurIPS does not allow us to share the revised manuscript itself at this stage of review.  Again, many thanks for pushing us in this valuable direction!

---

### Official Review · Reviewer_Wq9E · 2021-07-12

**Rating:** 6
**Confidence:** 3

**Summary:**

This paper proposes a very interesting framework of grammar and lexicon learning where lexical entries are grounded with visual image question answering.  The experiments are conducted and show the comparative performance of the proposed method to the state-of-the-art baselines.

**Limitations And Societal Impact:**

Please start with defining the primitives or initial representations of words and images.  It is not clear what are initially defined and given.

What about the scalability of the proposed method?  If it handles a small world with some objects, it will be fine.  However, there can be a number of different objects and regions and many properties and relations between them.  Is the proposed method feasible to such a more realistic situation?

**Main Review:**

The overall idea looks to be quite interesting, and the experimental results show the effectiveness of the proposal.
On the other hand, there are a number of points that are difficult to understand.

- How are the groundings between the image data and language expression provided?  Or, in what way the visual information is encoded and given?  Are the objects in the image represented by vectors as well?  How are their initial representations given?

- What are the set of primitives in the domain-specific language?  It looks "filter", "relate", and "query" are some of them.  Are there any other semantic functions?  Is such a set of primitive functions define for each specific task or domain?

- There are some lexical entries corresponding to the functional words such as prepositions and determiners.  How are they handled in the proposed framework?  While they do not have semantic information, they may have important syntactic information and are usually represented by complex category types.  It was a bit puzzling that such words are neglected in Figure 2 and still the derivation trees are constructed.

- It is not clear how the initial values are assigned to the lexical entries.

- Figure 4 is difficult to understand.  It is not clear if the gray parts in two candidates cover the same span in a given sentence.  More explanation is necessary.



**Time Spent Reviewing:**

6 hours

---

> ### Author Response · Authors · 2021-08-10
> **Author Response to Reviewer Wq9E**
>
> Thank you for your helpful comments.
>
> **Q1**: Image encoding on the CLEVR dataset.
>
> **A1**: Given the input image, we first use a pre-trained Mask R-CNN to detect object proposals, in the form of bounding boxes. We use a randomly initialized ResNet-34 model and the RoI Align layer to extract a vector representation for each object proposal. This ResNet-34 model as well as the concept embeddings, the grammar weights are jointly learned.
>
> ---
>
> **Q2**: The grounding between image data and language expression.
>
> **A2**: The grounding between vision and language is at the concept lever, that is, the visual-semantic embedding module. For example, in order to classify an object being shiny or not, we will compare the distance between the concept embedding of “shiny” and the vector embedding of the object. We do not use any explicit annotation for training this grounding. Instead, the model jointly learns the visual encoder (ResNet-34) and the concept embeddings by looking at images and reading paired questions and answers.
>
> ---
>
> **Q3**: Handling functional words in G2L2.
>
> **A3**: Since our DSL does not support representing all semantic programs of words, we allow certain words to be associated with an “empty” lexicon entry. This entry can be combined with any words or constituents next to it and does not participate in the composition of syntactic types and semantic programs. For example, in CLEVR, we do not explicitly model the meaning representation of be-words.
>
> ---
>
> **Q4**: Initial value assigned to lexical entries.
>
> **A4**: We have no prior about the lexical entry weights. Initially, the weights for all candidate lexical entries associated with all words in the vocabulary is 0. That is, for each word, we have a uniform distribution over the lexical entry associated with it.
>
> ---
>
> **Q5**: Figure 4.
>
> **A5**: Figure 4 is showing two candidate programs that can be potentially derived from the span: “right of the green cube” due to different lexical entry assignment. Specifically,
> - a lexical entry assignment leading to candidate 1 is: right - `N\N/N` `\lambda x,y.relate(y,x,RIGHT)`, green - `N` `filter(scene(),GREEN)`
> - a lexical entry assignment leading to candidate 1 is: right - `N\N/N` `\lambda x,y.relate(y,x,RIGHT)`, green - `N\N/N` `\lambda x,y.relate(y,x,GREEN)`, the - `N` `filter(scene(), THE)`, object - `N` `filter(scene(), OBJECT)`
>
> There are many other possible assignments which may lead to other candidate programs, because initially, we have no a priori knowledge about the lexicon: the word “green” can be a noun (`N`), an adjective (`N/N`), a preposition (`N\N/N`), etc.
>
> ---
>
> **Q6**: Scalability.
>
> **A6**: This paper focuses on grammar-based approaches for learning compositional and grounded representations. In the NS-CL paper, the authors have demonstrated generalizing this approach to learn visual concepts from natural images and sentences, but based on a pre-trained language parser, because the languages are more free-formed. We leave the extension to free form languages as future work.

---

### Official Review · Reviewer_MH2M · 2021-07-18

**Rating:** 5
**Confidence:** 3

**Summary:**

The paper presents a lexicon learning approach that learns to associate lexical items with "neural embeddings".
It does this by exploiting a given CCG grammar and learning the lexical itams
Results are tested on a visual reasoning and a language-driven navigation task
The main contribution of the paper is a specific joint parsing and execution algorithm that helps with training times.


**Limitations And Societal Impact:**

YES

**Main Review:**


Strength

Combining traditional parser with neural grounding is a very interesting research topic.

The approach is quite successful compared to baselines in the navigation domain and on some generalization tests in the visual reasoning domain.


Weaknesses

The parser is given in this system. I don't think that is the case for baselines MAC, NS-VQA, NS-CL. That makes the comparison to prior work somewhat weak. As you essentially add very strong biases with the given parser.
It then is somewhat surprising that the performance of the system is not so convincing compared to baselines except for generalization in the visual reasoning domain

In the visual reasoning domain - authors demonstrate that the approach is superior to baselines primarily with respect to depth. However, I think that is almost obvious since the parser should be doing a lot of the work here and that is given as a prior.

It's not clear what is the impact of the curriculum



Comments

L79 "to to"

**Time Spent Reviewing:**

2

---

> ### Author Response · Authors · 2021-08-10
> **Author Response to Reviewer MH2M**
>
> Thank you for your helpful comments.
>
> **Q1**: Parser is given to the system, giving it a very strong bias, yet the performance of the system is still not so convincing.
>
> **A1**: What’s given to G2L2 is a substantial bias of the parsing formalism, but it is very simple – three abstract rules that are candidates for universal generalizations about linguistic structure (see Figure 1 in the paper) – and the space of possible learned grammars is enormous, This, on the one hand, allows us to learn in very different datasets (CLEVR and SCAN), but on the other hand poses a serious learning challenge. A key contribution of our work is a successful learning algorithm that works on this space of grammars.
> We also want to point out that the performance of the system is very convincing. Please refer to our general response.
>
> ---
>
> **Q2**: Impact of curriculum learning.
>
> **A2**: There are two modules that could benefit from the curriculum learning setup: the language module --- learn word meaning from short sentences, and the visual concept learning --- learn concept embeddings from simple questions.
>
> To study how curriculum learning help these two modules, we conduct the following ablation study for the curriculum learning:
> The full G2L2 model trained with curriculum,
> The full G2L2 model trained without curriculum,
> The concept learning module trained with curriculum, while the grammar weights are copied and fixed from another trained G2L2 model,
> The concept learning module trained without curriculum, while the grammar weights are copied and fixed from another trained G2L2 model,
> Note that we can not fix the concept module but only train the grammar module because we don’t know the concept vocabulary.
>
> Due to the time limit, we haven’t seen the convergence of all models. We will add the full results when we have them. So far, our primary finding is that curriculum learning is important for convergence speed. Quantitatively, the convergence time of setting 4 is two times slower than setting 3.

---

> > ### Author Response · Authors · 2021-08-25
> > **Updates on the curriculum learning ablation study; looking forward to your feedback**
> >
> > We would like to update our results and findings regarding the ablation study on curriculum learning. Recall that we have four models.
> >
> > - A: The full G2L2 model trained with curriculum
> > - B: The full G2L2 model trained without curriculum
> > - C: The concept learning module trained with curriculum, while the grammar weights are copied and fixed from another trained G2L2 model
> > - D: The concept learning module trained without curriculum, while the grammar weights are copied and fixed from another trained G2L2 model
> >
> > Our primary findings have two parts:
> >
> > First, training all these models (A, B, C, D) on the 100% CLEVR images reach a similar final performance in terms of accuracy. However, compared with their counterparts without curriculum learning, the models with curriculum learning learn approximately 1.5x faster, in terms of the number of iterations. This is consistent with previous analysis of a similar model NS-CL on the same task [1].
> >
> > Second, interestingly, we see a statistical difference between the performance of A vs. B and C vs. D, when trained on 10% CLEVR images. Specifically, we have A: 98.11, B: 95.68, C: 98.15, D: 95.38.  Across all experiments, the training accuracy is ~99.9%. This suggests that curriculum learning helps prevent models from overfitting when the training data is small. Such overfitting happens mostly in the grounding of concepts (i.e., the concept learning part, by comparing C-D), instead of the grammar learning part. This is intuitive because when the scenes and questions get more complex, the model also gets more chance to find a spurious grounding of concepts that still yield the correct answer.
> >
> > We will include these findings in our paper. Our results suggest the following conclusions and future directions.
> > - Curriculum learning is helpful for concept learning, especially when the data is limited.
> > - There are simple ways to create curriculum splits. In this paper, we simply gradually add longer sentences into the training dataset.
> > - Future work can integrate our framework with more advanced protocols of curriculum learning, such as a competence-aware curriculum [1].
> >
> > ----------------
> >
> > Finally, we would like to thank you again for your reviews, comments, and suggestions. We sincerely hope these answers have addressed your questions. As the end of the discussion period is approaching, we want to be sure we have addressed all of your concerns.  Please let us know if you have any additional comments.
> >
> > ----------------
> >
> > **References**
> >
> > [1] Qing Li, Siyuan Huang, Yining Hong, Song-Chun Zhu. A Competence-aware Curriculum for Visual Concepts Learning via Question Answering. In ECCV 2020.

---

### Official Review · Reviewer_JDyB · 2021-07-20

**Rating:** 7
**Confidence:** 4

**Summary:**

This paper proposes a fusion of CCG-based approach and concept embedding modeling to introduce a neuro-symbolic approach for grounded lexicon learning. The experiments are done on closed-world and relatively closed-language datasets to empirically test the effectiveness of the proposed method, especially on compositional generalization. Based upon a lexicon of types and attributes described in a task specific DSL (Domain Specific Language), and using the training signal from downstream task (visual QA and instruction parsing), this approach learns concept embeddings and syntactic roles of types in surface form of text. The CCG parse is used to define ann executable program and the program itself requires concept embeddings for execution.

A fairly straightforward modification to the standard CKY algorithm, based upon subtree merging of same types is proposed to compute the derivations and weights in a tractable manner. The results show improvements over relevant baselines in terms of length and syntactic depth generalization.

**Limitations And Societal Impact:**

Yes, they have sufficiently addressed broader societal impact of their work.

**Main Review:**

Positives:
-- The paper for the most part is clearly written with ample motivation and example (except the CKY section as mentioned below).

-- The empirical results convincingly show the effectiveness of the proposed approach on the nested-structure and length generalization.

-- The baselines are relevant and the two sufficiently dissimilar tasks strengthen the claims about generalizability of the proposed approach.

Negatives:
-- The method seems difficult to scale to free-form language in an open-world setting. Access to a DSL is assumed an the method does rely on types, attributes, and program definitions supported in the DSL. The approach is tested on limited vocabulary which is further pruned to eliminate lexical ambiguity and only preserve syntactic ambiguity. However, with such a closed-world/language setup, the challenges associated with processing free-form natural language are not addressed.

-- The CKY section is difficult to read. Derivation_k is not explained anywhere. What is k here? Similarly, the first equation in the section does not seem correct--either it is a very strong assumption, or the equation is poorly written/explained. I checked the appendix and the equations in the appendix make more sense regarding \exp and \sum.

-- I am still confused about the impact of expected execution. Don't we need the highest scoring derivation (or top-k derivations), both during learning (as described in the loss), and inference. The expected quantities involving marginalization do not appear in either the loss or decoding objectives. It appears that the method is a hybrid of enumeration of derivations and computing expected quantities (since this execution is limited to particular subtrees). The impact of this modeling choice should be discussed in more detail.

-- In some settings, the proposed approach trails other baselines on the CLEVR dataset. Is there a systematic reason for this? Or can it just be attributed to unsuccessful grammar induction?

**Time Spent Reviewing:**

7

---

> ### Author Response · Authors · 2021-08-10
> **Author Response to Reviewer JDyB**
>
> Thank you for your helpful comments.
>
> **Q1**: Scale to free-form language.
>
> **A1**: We agree that handling free-form language that includes sense ambiguities is a meaningful direction. G2L2 can potentially be extended to this setting. One straightforward idea is to use neural networks (e.g., RNNs) to encode contextual information and use neural networks to predict the lexical entry for each word in a context. We have tried this idea but found it having serious issues in the generalization aspect. Specifically, in the SCAN domain, the lexicon for “right” is: `lambda x. concat(TURN_RIGHT, x)` when used with “look” and “walk” (prepend a right turn operation as in `look right = TURN_RIGHT LOOK`) but `lambda x: concat(x, RUN)` when used in “run right”. The semantics of “run” in “run right” is `TURN_RIGHT`. This perfectly fits the training set, but leads to poor generalization to “jump right”.
>
> There could be designs such as adding regularizations for “word meanings tend to be consistent across contexts”, and mechanisms for determining when to add new senses to words during learning. We see these as important challenges and believe models and benchmarks should be proposed for systematically studying this challenge.
>
> ---
>
> **Q2**: Access to DSL.
>
> **A2**: Our work is constrained in the sense that we only support a limited set of types and operations defined over these types: object properties, relations, boolean values and integers. But it is also general because we do not rely on a predetermined set of concepts (red, cube) or attributes (color, shape).
>
> ---
>
> **Q3**: Denotations in CKY: definition of derivations, first equation.
>
> **A3**: In the notation $derivation_k$, $k$ is an index for candidate derivations, where $k=1,2,...M$, where $M$ is the total number of derivations parsed from the sentence, because different words can take different lexicon entries.
>
> The first equation in the CKY section denotes the context-independent assumption of lexicon weights: the lexicon entry associated with each word is independent with respect to its context. See also our discussion about handling free-form language.
>
> ---
>
> **Q4**: The impact of the expected execution.
>
> **A4**: Regarding the loss function. We first compute the expected execution result of all candidate derivations using CKY-E2. Then we compute the loss between the groundtruth answer and this expected execution result.
>
> More concretely, in the loss function, we want to compute the loss regarding every possible derivation generated from the sentence (not just top K). If we reuse the notation from A3 above, the sum over $k$ should be written as sum from $k=1$ to $k=M$.
>
> Thus, expected execution is important to avoid computing an exponential number of possible derivations. For example, in the SCAN domain, the longest sentence with 9 words can has 10^14 possible derivations based on our candidate lexicon set. This is impossible for the system to compute and do gradient descent over. We also have a detailed proof about the complexity of CKY-E2 in Section 2.2 in the supplementary.
>
> ---
>
> **Q5**: Performance on CLEVR.
>
> **A5**: G2L2 sometimes fails to output the correct program for certain sentences. One failure mode we find in G2L2 is the phrase “in front of”. In certain runes, it associates the word “in” with a conjunction “or” and the word “of” with an object filter. The question “how many cubes are in front of the sphere?” will yield `query(union(filter(cube), filter(of, filter(sphere))))`. Although it achieves almost perfect accuracy (>99.5%) on the training set, such program structure fails to generalize to test images. Note that NS-CL does not have this issue because it has access to the concept vocabulary. That is, G2L2 is solving a harder problem.

---

> > ### Comment · Reviewer_JDyB · 2021-08-25
> > **Thanks for the author response**
> >
> > Thanks for responding to my review!

---

### Author Response · Authors · 2021-08-10
**General Author Response**

We thank all reviewers for their helpful comments and suggestions. We are happy to see that reviewers generally found our approach to be important and novel.

Based on multiple reviewer comments, we also recognize that our results section needs revision to more clearly explain our findings and the value added by our G2L2 model over baselines.  To summarize three points we will bring out in revision:

(1) On both CLEVR and SCAN datasets, G2L2’s learned grammar-based lexicon **allowed more sample-efficient learning** than earlier approaches that do not use explicit symbolic representations of concepts (MAC, TbD-Net) in CLEVR, or approaches that do not use dataset augmentation in SCAN (standard Seq2Seq models); the performance improvement in CLEVR was substantial.

(2) Our approach also gave **much stronger out-of-sample generalization**.  In SCAN, the original dataset comes with several tests of compositional generalization (holding out concepts “jump” and “around right” from training except for very short sentences) and length-based generalization (holding out longer sentences from training). In CLEVR, we developed new train/test splits to test out-of-sample generalization to sentences with more complex compositional structure (holding out concepts “purple”, “right-of” and “count” from training except for very short sentences) and greater embedding depth.  Across each of  these tests, some baseline models performed well and others poorly, but in both domains, **our model was the only one that solved all these generalization tasks**.

(3) Most importantly and revealingly, our model was **the only one to perform well on the hardest out-of-sample generalization tests**: holding out “count” in CLEVR, and generalizing to deeper embeddings in CLEVR and longer utterances in SCAN.  The other, easier generalization tests all have close neighbors in the training set, differing by just one word.  In contrast, the length, depth and “count” tests require generalizing to sentences that differ in multiple words from any training example. They appear to require -- or at least benefit especially well from -- G2L2’s lexical-grammatical approach to capturing meaning of complex utterances, with explicit constituent-level (as opposed to simply word-level) composition.

Taken together, we believe these results clearly show the value of the key contribution in G2L2: integrating a learnable grammar-based lexical representation into grounded neuro-symbolic language models.  We elaborate on these points in response to specific reviewer comments below, but we feel it is important to highlight them here for all reviewers and the meta-reviewer. We also address each reviewer’s individual critiques below.  All of the points raised were good ones and we feel the paper will become much stronger by incorporating these responses into our revised final paper.

---

### Decision · Program_Chairs · 2021-09-28

**Decision:**

Accept (Poster)

**Comment:**

This paper presents a novel CCG-based neuro-symbolic framework for grounded language tasks like VQA. Experiments are conducted on both VQA (CLEVR) and navigation tasks. Results demonstrate marked improvements in generalization to particularly challenging question types (e.g. counting in CLEVR). Overall, reviewers are borderline -- half in favor of acceptance, half weakly against.    Several reviewers praised the paper's clarity and well-articulated motivations. Several also viewed the demonstrated improvements to generalization as impactful, and found the proposed training method (which helps address computational costs associated with latent derivations) to be a valuable contribution. However, reviewers also raised several important concerns. First, one reviewer raised concerns about the potential rigidity of the proposed formalism (e.g. that aspects of the grammar must be specified in advance) -- though authors have pointed out how simple and generalizable (e.g. between VQA and navigation tasks) the pre-specified component is. Second, one reviewer was concerned that the main hypothesis of the paper -- that taking a lexicalist approach to grounded language tasks may offer benefits -- could be better motivated, analyzed, and justified. The same reviewer raised a broader, but related point about the readability of the current draft. The draft, while clear, is quite dense and depends to some extent on both strong familiarity with deep learning and specific branches of linguistics, potentially limiting its audience and impact. Taking these points in balance, I lean towards acceptance. The paper does present a novel and well-motivated approach that may, apart from the this specific application, have influence on others working with models that combine discrete syntactic formalisms with neural representations. The paper does demonstrate strong improvements in generalization to unseen question types, matching the initial motivation of the general shape of approach. However, I strongly agree that final revisions should include a better survey and introduction to the relevant linguistic concepts and, perhaps even more importantly, more in depth discussion of how these specific results relate to the lexicalist hypothesis, around which the framework is centered.

**Consistency Experiment:**

NeurIPS has a long history of experimentation. In 2014, NeurIPS ran an experiment in which 10% of submissions were reviewed by two independent committees to quantify the randomness in the review process. This year, we repeated a variant of this experiment to see how the quality of the review process has changed over time.  This paper was part of the experiment and was therefore assigned to two committees (consisting of reviewers, an Area Chair, and a Senior Area Chair) that reached independent decisions.  If both committees made the same recommendation, this recommendation was followed. If a single committee recommended acceptance, the paper was accepted (with the exception of a few cases in which the other committee identified what we considered a fatal flaw, e.g., an error in a key result).

Both committees reached the same decision: **Accept (Poster)**

The other committee assigned to the paper recommended **Accept (Poster)**.  You can find the other set of reviews, along with any follow up discussion with the authors here:
https://openreview.net/forum?id=iI6nkEZkOl